# PlantRSR: A New Plant Dataset and Method for Reference-based Super-Resolution

**Hongyang Zhou**[1], **Xiaobin Zhu**[1,*] **Shengxiang Yu**[2*], **Liuling Chen**[1],
**Jingyan Qin**[1], **Xu-Cheng Yin**[1]
[1]University Of Science and Technology Beijing,
[2]Institute of Botany, Chinese Academy of Sciences

## Abstract

Single image super-resolution (SISR) often struggles to reconstruct high-resolution (HR) details from heavily degraded low-resolution (LR) inputs. Instead, reference-based super-resolution (RefSR) methods offer an alternative solution to generate promising results using high-quality reference (Ref) images to guide reconstruction. However, existing RefSR datasets focus on limited scene types, primarily featuring human activities and architectural scenes. Plant scenes exhibit complex textures and fine details, essential for advancing RefSR in natural and highly detailed scenes. To this end, we meticulously captured and manually selected high-quality images containing rich textures to construct a large-scale plant dataset, **PlantRSR**, comprising 16,585 HR–Ref pairs. The dataset captures the complexity and variability of plant scenes through extensive variations. In addition, we propose a novel RefSR method specifically designed to tackle the distinct challenges posed by plant imagery. It incorporates a Selective Key-Region Matching (SKRM) that selectively identifies and performs matching between LR and Ref images, focusing on distinctive botanical textures to improve matching efficiency. Additionally, a Texture-Guided Diffusion Module (TGDM) is proposed to refine LR textures by leveraging a diffusion process conditioned on the matched Ref textures. TGDM is effective in modeling irregular and fine textures, thereby facilitating more accurate SR results. The proposed method achieves significant improvements over state-of-the-art (SOTA) approaches on our PlantRSR dataset and other benchmarks. Code and dataset are released at: `https://github.com/edbca/PlantRSR`.

## 1 Introduction

SISR aims to reconstruct an HR image from an LR input, and has been widely applied in various fields, including critical uses in plant phenotyping, where high-resolution imagery is crucial for detailed morphological analysis, disease diagnosis, and growth monitoring in precision agriculture. While SISR has shown promising results Dai et al. (2019); Niu et al. (2020); Huang et al. (2021); Kong et al. (2021); Wang et al. (2021); Zhou et al. (2023; 2026); Wang et al. (2024a); Guo et al. (2025), it often fails to reconstruct precisely detailed textures of the ground-truth HR image when the original high-frequency information is severely lost during the degradation. To address this issue, RefSR methods Zhang et al. (2019); Yang et al. (2020); Lu et al. (2021); Jiang et al. (2021); Xia et al. (2022); Cao et al. (2022); Zhou et al. (2025) incorporates external Ref images as guidance, leveraging external high-quality textural details to enhance texture recovery.

Recent RefSR methods leverage feature alignment, attention mechanisms, or implicit correspondence learning to fuse textures from Ref images. Despite significant progress, existing studies primarily focus on limited scenes such as human daily life and architectural structures, as reflected by datasets like CUFED5 Zhang et al. (2019) and LMR Zhang et al. (2023). As shown in Fig. 1 (a) and (b), CUFED5 and LMR mainly contain scenes with rigid structures and limited geometric variations. In contrast, plant scenes, as shown in Fig. 1 (c), exhibit distinct characteristics such as significant shape deformation, subtle texture variations, and frequent defocused backgrounds, which introduce

---

*Xiaobin Zhu and Shengxiang Yu are the corresponding authors.

additional challenges for RefSR. Moreover, the lack of publicly available RefSR datasets tailored to plant imagery restricts the development of methods capable of handling its unique challenges. Although the DRefSR Zhou et al. (2025) expands upon CUFED5 by increasing data diversity and includes certain plant images, it is not specifically designed for plant scenes and suffers from insufficient quantity and variety. As a result, existing approaches often struggle to generalize to such complex and diverse scenarios. In particular, plant imagery presents unique challenges, such as irregular structures and diverse textures, that differ significantly from the relatively structured content found in current datasets. This highlights the necessity for RefSR datasets and strategies specifically tailored to plant imagery.

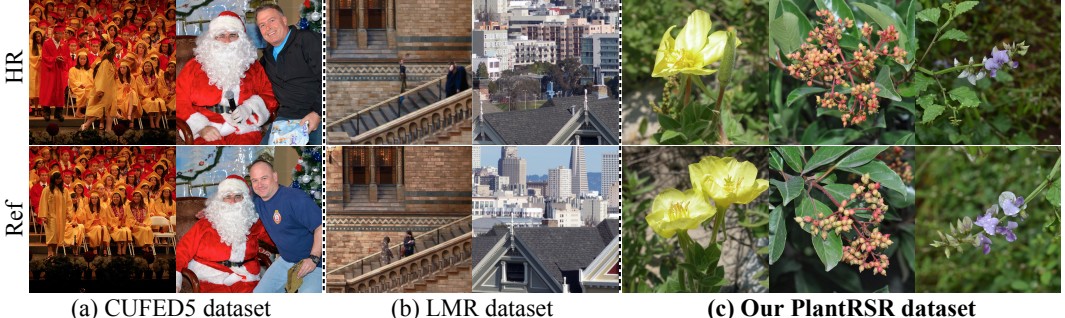

| (a) CUFED5 dataset | (b) LMR dataset | (c) Our PlantRSR dataset |

Figure 1: Example image pairs from different RefSR datasets. (a) CUFED5 dataset Zhang et al. (2019), primarily featuring scenes of human daily activities. (b) LMR dataset Zhang et al. (2023), focused on architectural scenes. (c) Our PlantRSR dataset, designed for complex botanical textures with diverse plant species.

To address the above challenges, we construct a novel large-scale RefSR dataset specifically designed for plant imagery, named **PlantRSR**. All images were carefully captured by us from real-world plant scenes, ensuring high authenticity and diversity. The dataset comprises 6,134 high-quality HR–Ref image pairs with extensive variations in color, rotation, deformation, and background blur, covering resolutions from 2K to 8K. In RefSR tasks, models are commonly trained using cropped image patches, since full-resolution images are often too large for direct processing. However, generating aligned patch pairs is particularly challenging. Automated patch generation in previous datasets often leads to misaligned or semantically inconsistent pairs, which is unsuitable for complex plant imagery. To ensure high-quality training data, we manually annotate and construct 16,585 semantically aligned HR–Ref patch pairs, exceeding the 11,817 pairs in the widely used CUFED5 dataset. Besides, the PlantRSR dataset includes a testing set of 100 image pairs, offering the first RefSR benchmark for plant imagery and a solid foundation for future research.

We introduce an innovative RefSR approach tailored to address the distinct challenges in plant environments. First, we propose a Selective Key-Region Matching (SKRM) that can target the focal characteristics of plant images, performing matching only on key regions between LR and Ref images to significantly enhance matching efficiency. Second, we present a Texture-Guided Diffusion Module (TGDM), which enhances LR features by incorporating matched Ref textures as conditional guidance within a diffusion-based refinement framework. This novel module effectively leverages the fine-grained and irregular textures from the Ref image to improve the quality of the SR image. Together, these components form a unified framework that ensures both accurate texture transfer and efficient feature enhancement. Extensive experiments demonstrate the superiority of our method. In summary, our main contributions are four-fold:

- We contribute a new large-scale plant dataset for RefSR, named PlantRSR, which contains 16,585 high-quality training pairs with rich variations in color, rotation, deformation, and background blur, reflecting the complexity of real-world plant scenes.
- We propose a novel RefSR method specifically designed to tackle the challenges posed by plant imagery. Extensive experiments on multiple datasets demonstrate that our method outperforms SOTA methods both quantitatively and qualitatively.
- We propose a Selective Key-Region Matching (SKRM) module that leverages the focal characteristics of plant imagery to perform region-specific matching between LR and Ref images, thereby significantly enhancing matching effectiveness.

- We propose a Texture-Guided Diffusion Module (TGDM) that introduces a diffusion mechanism conditioned on matched Ref textures to refine LR regions, enabling more accurate reconstruction of the SR image.

## 2 RELATED WORK

**Reference-based Image Super-Resolution.** Unlike conventional SISR methods that rely exclusively on the LR input, RefSR incorporates an additional high-quality Ref image to supply textures and structural details. Pioneer RefSR methods mainly emphasized alignment between the LR and Ref images. For instance, Wang *et al.* Wang et al. (2016b) proposed iterative non-uniform warping to refine the Ref for better texture extraction. CrossNet Zheng et al. (2018) employs multi-scale optical flow to achieve spatial alignment, while SSEN Shim et al. (2020) adopts deformable convolutions (Dai et al., 2017; Zhu et al., 2019) to align feature representations adaptively.

In addition to alignment-based strategies, a significant body of work has explored patch-based texture transfer. SRNTT Zhang et al. (2019), for example, utilizes perceptual features extracted from VGG Simonyan & Zisserman (2015) to identify semantically similar textures between LR and Ref images. E2ENT2 Xie et al. (2020) proposes a task-specific feature extraction framework, moving beyond general classification features. TTSR Yang et al. (2020) designs a cross-scale transformer to effectively aggregate textures across multiple levels. Other approaches, such as AMRSR Pesavento et al. (2021) and CIMR-SR Yan et al. (2020), expand to multi-Ref settings for richer texture guidance. MASA Lu et al. (2021) addresses efficiency through a hierarchical coarse-to-fine matching scheme. To tackle domain discrepancies in resolution and content, $C^2$-Matching Jiang et al. (2021) introduces contrastive learning He et al. (2020) and knowledge distillation Hinton et al. (2015) for robust correspondence learning. At the same time, DATSR Cao et al. (2022) leverages transformer architectures for better handling of scale and transformation variations. RRSR Zhang et al. (2022) proposes a reciprocal mechanism where reconstruction feedback enhances the overall performance. Ref-IRT Zhang et al. (2024) introduces a progressive restoration pipeline for handling complex degradations. MCMSR Zheng et al. (2024) focuses on discovering multiple matching candidates in the Ref image for each LR region to improve completeness. SSMTF Zhou et al. (2025) attempts to leverage state-space models to extract multi-scale information from Ref images. Recently, diffusion models have emerged as a powerful paradigm for leveraging Ref information. DiffMSR Li et al. (2024a) employs diffusion to generate prior knowledge from Ref for magnetic resonance image reconstruction. Similarly, in remote sensing, Ref-Diff Dong et al. (2024) proposes a change-aware diffusion model to extract change priors from Ref images. CoSeR Sun et al. (2024), instead of using a physical Ref, utilizes a diffusion model to generate a semantically relevant Ref image to aid reconstruction. This idea of *generating* references connects to retrieval-augmented methods; for example, RAG Lee et al. (2025) proposes a Retrieval-Augmented Generation framework that retrieves a Ref image given an LR query. However, most existing methods still face limitations in complex scenarios like plant imagery, which features irregular structures and diverse textures that are significantly different from the current dataset.

**Datasets for Reference-based Image Super-Resolution.** SISR datasets such as DIV2K Timofte et al. (2017) and Flickr2K Lim et al. (2017) have been widely adopted to learn mappings from LR to HR images. However, they are not well-suited for RefSR tasks, as they lack external Ref images that are crucial for guiding texture enhancement Zheng et al. (2018); Zhang et al. (2019). To bridge this gap, CUFED5 Zhang et al. (2019) is proposed as an early dataset tailored for RefSR, offering 13,761 training pairs of size 160×160, each consisting of a HR image and a corresponding Ref image. CUFED5 provides 126 test groups for evaluation, where each group includes an HR image and four Ref images with varying degrees of similarity. Following this, Lin *et al.* Zhang et al. (2023) introduced the LMR dataset, a large-scale multi-reference benchmark designed to further enrich RefSR research. It comprises 112,142 training groups of 300×300 images, each paired with five Ref images across different similarity levels. The testing set includes 142 groups, where each target image is matched with 2 to 6 Ref images. To address the limited diversity of CUFED5, DRefSR Zhou et al. (2025) extends this dataset by constructing a more varied training set comprising 13,761 image pairs. In addition, other efforts, such as Sun80 Sun & Hays (2012) and WR-SR Jiang et al. (2021) offer testing scenarios more aligned with real-world applications. Sun80 provides 80 natural images with internet-sourced reference sets, while WR-SR contains 100 image pairs featuring references retrieved through web search, encompassing diverse visual categories such as animals and landmarks.

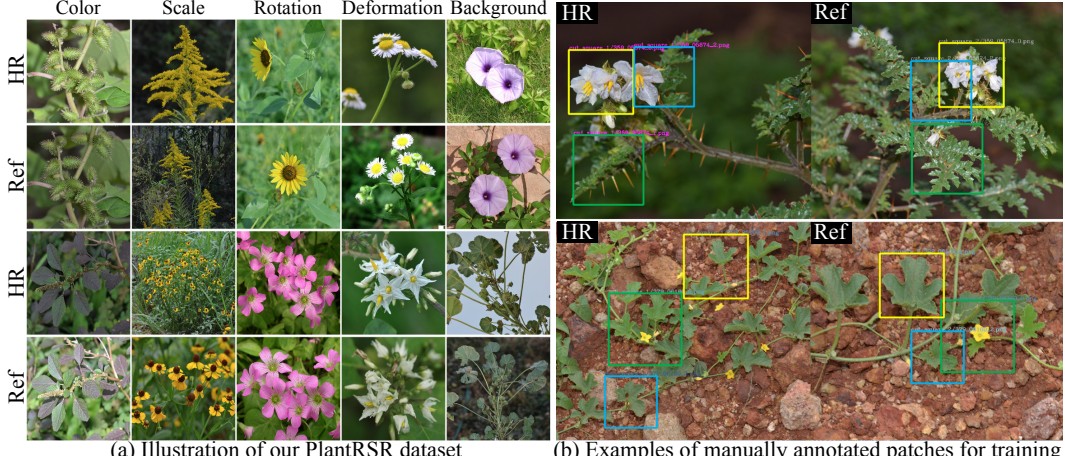

(a) Illustration of our PlantRSR dataset      (b) Examples of manually annotated patches for training

Figure 2: Category overview and training set construction of our PlantRSR dataset.

Despite recent progress, existing RefSR datasets like CUFED5 and LMR offer limited diversity and are ill-suited for plant phenotyping. Although DRefSR expands the variety of scenes and includes some plant imagery, the quantity and diversity of plant species remain insufficient (Detailed analysis in Appendix C). Plant images pose unique challenges such as large shape deformations, subtle textures, and frequent defocus, which are not well represented in current datasets.

# 3 PLANTRSR DATASET CONSTRUCTION

The widely used CUFED5 dataset is constructed from CUFED Wang et al. (2016a) and primarily targets human daily life scenes. Similarly, the LMR dataset, built upon MegaDepth Li & Snavely (2018), focuses mainly on landmark scenarios. These datasets are not well-suited for plant scenes. Furthermore, existing plant-specific datasets remain scarce, making it challenging to obtain valid HR-Ref image pairs. To meet the needs of current RefSR tasks, we meticulously captured 6,134 image pairs with rich textures using DSLR cameras. The images cover resolutions between 2K and 8K, over 80% of which are above 4K. To ensure the diversity and realism of the dataset, we deliberately captured images that reflect five major types of variations commonly encountered in real-world plant photography: color variations (11.4%), scale differences (18.6%), rotations (27.0%), deformations (30.5%), and background changes (12.5%). These variations are introduced by adjusting camera angles, distances, and viewpoints during data collection. Fig. 2 (a) provides visual examples of each category, illustrating the challenges posed by different reference conditions in the plant domain.

In SISR, it is common to crop patches from HR images for training randomly. However, in RefSR, corresponding regions with similar content must be cropped from both the HR and Ref images. For example, CUFED5 first randomly crops 160×160 patches from HR images and then selects corresponding patches from Ref images based on predefined correspondence. Similarly, LMR randomly selects a 300×300 patch from the HR image, projects its center to a sparse 3D point cloud, and retrieves Ref patches centered at nearby keypoints. However, these automated cropping strategies are not suitable for plant images. Due to the frequent presence of defocused backgrounds and complex structures in plant scenes, random or keypoint-based cropping often results in uninformative or mismatched patches. Moreover, the significant variations in our paired plant images, such as color, scale, and deformation differences, make accurate automatic correspondence challenging. Therefore, we adopt a manual annotation strategy to carefully select semantically matched regions from image pairs, as illustrated in Fig. 2 (b). Then, we obtain 16,585 patch pairs, which are rescaled into two versions (160×160 and 300×300) to construct the training set (More samples in Appendix K).

# 4 OUR METHOD

The overview of our method is shown in Fig. 3 (a). Given the LR image $I^{LR}$ and the Ref image $I^{Ref}$, we aim at generating a high-quality image $I^{SR}$ that possesses texture-rich details conditioned on $I^{Ref}$. For correspondence matching, we follow $C^2$-Matching Jiang et al. (2021) and adopt

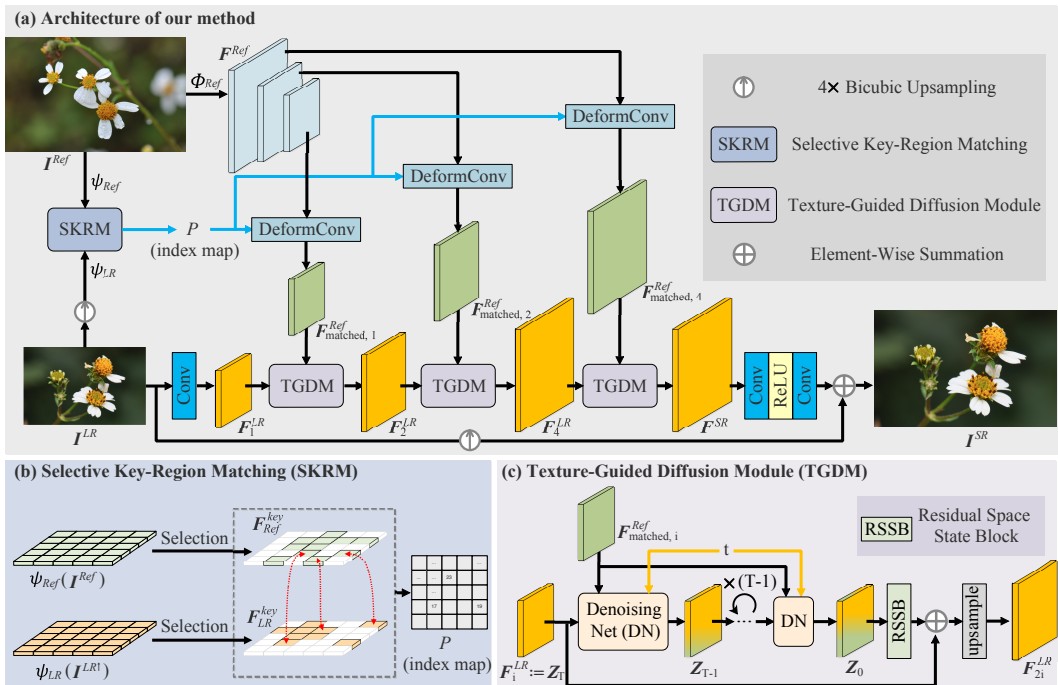

Figure 3: (a) Architecture of our method. (b) Illustration of SKRM. (c) Illustration of TGDM.

contrastive learning He et al. (2020) and knowledge distillation Hinton et al. (2015) to train the Ref feature extractor $\psi_{Ref}$ and the LR feature extractor $\psi_{LR}$. These extractors are used to obtain $\psi_{Ref}(I^{Ref})$ and $\psi_{LR}(I^{LR\uparrow})$ from the Ref and upsampled LR images, respectively. Afterward, our proposed SKRM selectively matches key regions between $\psi_{Ref}(I^{Ref})$ and $\psi_{LR}(I^{LR\uparrow})$ by focusing on informative areas, thereby improving matching efficiency and obtaining the correspondence index map $P$. To utilize both high-level and low-level information provided by $I^{Ref}$, following previous works Jiang et al. (2021); Xia et al. (2022); Cao et al. (2022); Zhang et al. (2022), we extract Ref texture features $F^{Ref}$ at different scale levels via encoder $\Phi_{Ref}$ Simonyan & Zisserman (2015). At each scale, we perform the deformable convolution networks Dai et al. (2017); Zhu et al. (2019) on $F^{Ref}$ to warp it to match the $F^{LR}$ for later fusion. Finally, our proposed TGDM leverages the diffusion process conditioned on $F^{Ref}_{matched}$ to progressively enhance $F^{LR}$, thereby enabling more effective multi-scale feature fusion and producing the final SR image $I^{SR}$. We employ the L1 loss function as the optimization objective during model training.

## 4.1 SELECTIVE KEY-REGION MATCHING (SKRM)

Since plant images often contain detailed foregrounds and shallow DoF-induced blurred backgrounds that are easy to reconstruct, explicit matching in such regions is unnecessary. Instead, as illustrated in Fig. 3 (b), we focus on selecting and matching key textured regions. To identify these regions, we introduce a key detail selection indicator. Given a feature map $F \in \mathbb{R}^{H \times W \times C}$ and a sampling ratio $s = 2$, the key detail selection metric $M_F \in \mathbb{R}^{H \times W}$ (values 0 or 1) is obtained through the operation $S(\cdot)$, defined as the absolute difference between the original feature map and its reconstruction via bilinear downsampling and upsampling as:

$$M_F = S(F) = \mathbb{I}(\sum_C | F - F_{\downarrow s \uparrow s} | > \tau), \tag{1}$$

where $\tau$ is the mean plus standard deviation of the absolute differences, $\mathbb{I}(\cdot)$ returns 1 when the input exceeds threshold $\tau$, and 0 otherwise. Considering that the Ref image may also contain blurred background regions, we apply key region selection to the Ref image to enhance matching efficiency with the LR image. Based on Eq. 1, the key detail selection indicators are derived as follows:

$$M_{Ref}, M_{LR} = S(\psi_{Ref}(I^{Ref})), S(\psi_{LR}(I^{LR\uparrow})). \tag{2}$$

Subsequently, the key texture features $F_{Ref}^{key}$ and $F_{LR}^{key}$ are then derived from these indicators as:

$$F_{Ref}^{key} = M_{Ref} * \psi_{Ref}(I^{Ref}), \tag{3}$$

$$F_{LR}^{key} = M_{LR} * \psi_{LR}(I^{LR\uparrow}), \tag{4}$$

Afterward, we get descriptors $d^{Ref} = [k_1, ..., k_m]$ and $d^{LR} = [q_1, ..., q_n]$ from $F_{Ref}^{key}$ and $F_{LR}^{key}$, respectively. These descriptors are obtained by folding extracted features into patches. Then we find the patch from the patches $k_j$ that is most similar to each LR patch $q_i$ as:

$$P_i = Top1(\frac{q_i}{\|q_i\|} \cdot \frac{k_j}{\|k_j\|}), \tag{5}$$

where $Top1(\cdot)$ is a function and return the most relevant positions $P_i$.

Then, we use a deformable convolution $DConv(\cdot)$ and the position index map to match the similar textures around every position $P_i$ of $F^{LR}$ as:

$$\begin{aligned} F_{\text{matched},l}^{Ref} &= DConv(F_l^{Ref}, P_i) \\ &= \sum wF_l^{Ref}(P + P_0 + P_i + \Delta p)\Delta m, \end{aligned} \tag{6}$$

where $F_l^{Ref}(l = 1, 2, 4)$ denotes Ref texture features at different scale via encoder $\Phi_{Ref}$, $P_0 \epsilon \{(-1,1), (-1,0), ..., (1,1)\}$, $w$ denotes the convolution kernel weight, $\Delta p$ and $\Delta m$ denote the learnable offset and modulation scalar, respectively.

## 4.2 TEXTURE-GUIDED DIFFUSION MODULE (TGDM)

As shown in Fig. 3 (c), TGDM enhances the LR feature $F_l^{LR}$ by leveraging the matched Ref texture $F_{\text{matched},l}^{Ref}$ as conditional guidance within a diffusion-based refinement framework, enabling a progressive refinement of the LR features with Ref guidance.

We denote the initial LR feature as $F_l^{LR} := Z_T$, which serves as the starting point for the reverse diffusion process. To recover $Z_0$ from $Z_T$, we adopt a conditional denoising process Ho et al. (2020); Li et al. (2024b). At each timestep $t \in \{T, T-1, ..., 1\}$, the Denoising Network (DN) predicts the noise $\hat{\epsilon}_\theta(Z_t, t, F_{\text{matched},l}^{Ref})$ in $Z_t$ conditioned on the matched Ref texture, and each sampling step can be expressed as:

$$q(Z_{t-1}|\hat{Z}_0, Z_t) = \mathcal{N}(Z_{t-1}; \tilde{\mu}_t(\hat{Z}_0, Z_t)\tilde{\beta}_t I), \tag{7}$$

$$\hat{Z}_0 = \frac{1}{\sqrt{\bar{\alpha}_t}} \left( Z_t - \sqrt{1-\bar{\alpha}_t} \cdot \hat{\epsilon}_\theta(Z_t, t, F_{\text{matched},l}^{Ref}) \right), \tag{8}$$

$$\tilde{\mu}_t(\hat{Z}_0, Z_t) = \frac{\sqrt{\bar{\alpha}_{t-1}}\beta_t}{1-\bar{\alpha}_t}\hat{Z}_0 + \frac{\sqrt{\alpha_t}(1-\bar{\alpha}_{t-1})}{1-\bar{\alpha}_t}Z_t, \tag{9}$$

$$\tilde{\beta}_t = \frac{1-\bar{\alpha}_{t-1}}{1-\bar{\alpha}_t}\beta_t, \tag{10}$$

where, $\alpha_t$ and $\beta_t$ are the forward process coefficients, and $\bar{\alpha}_t = \prod_{s=1}^t \alpha_s$. This denoising step is iteratively applied from $t = T$ to $t = 1$, producing the refined latent $\hat{Z}_0$ guided by the Ref texture. To further enhance textures feature, we pass $Z_0$ through a Residual State Space Block (RSSB) Guo et al. (2024) as:

$$\bar{Z}_0 = \text{RSSB}(Z_0). \tag{11}$$

The refined latent $\bar{Z}_0$ is then fused with the original LR feature via residual addition and subsequently upsampled using a sub-pixel convolution layer Shi et al. (2016) as:

$$F_{2l}^{LR} = \text{upsample}(F_l^{LR} + \bar{Z}_0), \quad l = 1, 2. \tag{12}$$

Finally, the enhanced feature $F_4^{LR}$ is used to generate the super-resolved representation by an additional residual connection as:

$$F^{SR} = F_4^{LR} + \bar{Z}_0. \tag{13}$$

TGDM progressively injects Ref-guided textures into the LR representation through the conditional diffusion mechanism, improving feature quality for high-fidelity SR reconstruction.

## 5 EXPERIMENTS

### 5.1 DATASETS AND IMPLEMENTATION DETAILS

Due to the unavailability of the LMR dataset, we employ CUFED5, DRefSR, and our PlantRSR training datasets for comparative evaluation. All models are trained on $160 \times 160$ patches under identical settings to CUFED5 for consistency. The LR images are produced by $4 \times$ bicubic downscaling of HR images. All SR results are evaluated using four metrics: PSNR, SSIM Wang et al. (2004), LPIPS Zhang et al. (2018), and DISTS Ding et al. (2020). The PSNR and SSIM values are calculated on the Y channel in YCbCr color space. Besides, we introduce M-PSNR, a variant that measures PSNR only in textured regions using a mask similar to Eq. 1. In our method, the number of $T$ is set to 4. We train our model with 400 epochs using ADAM optimizer ($\beta_1$=0.9 and $\beta_2$=0.999). The initial learning rate is set to $10^{-4}$ and decreased by 0.5 after each $100,000$ iteration. We implement experiments using the NVIDIA RTX A6000 GPU.

### 5.2 QUANTITATIVE EVALUATION

We compare our method with several SOTA RefSR approaches, including MASA Lu et al. (2021), $C^2$-Matching Jiang et al. (2021), AMSA Xia et al. (2022), DATSR Cao et al. (2022), RRSR Zhang et al. (2022), MRefSR Zhang et al. (2023), HiTSR Aslahishahri et al. (2024), MCMSR Zheng et al. (2024) and SSMTF Zhou et al. (2025). All comparison methods are trained independently on both the CUFED5 dataset and our PlantRSR dataset. The detailed experimental results are listed in Tab. 1. Our method achieves the best performance across all four metrics on all training datasets. Furthermore, experimental observations reveal that models trained on PlantRSR consistently outperform those trained on CUFED5 and DRefSR. Notably, **Ours** achieves superior performance with only 11.1M parameters, significantly fewer than competitors like RRSR (21.5M) and MRefSR (23.7M). Moreover, all methods exhibit consistent performance improvements when trained on PlantRSR compared to both CUFED5 and DRefSR. Besides, the significantly greater improvement in M-PSNR compared to standard PSNR demonstrates our dataset's enhanced effectiveness for complex plant textures. Experimental results validate method superiority and PlantRSR dataset significance.

### 5.3 QUALITATIVE EVALUATION

We visually compare our method and other methods in Fig. 4. In Fig. 4 (a-c), we present the visual comparison of plant leaf reconstruction. The restoration of leaf venation patterns proves particularly challenging, where our method demonstrates superior texture recovery compared to other approaches. Fig. 4 (d) and (e) present two challenging scenarios involving both color and scale variations. Despite these difficulties, our method achieves superior reconstruction quality compared to other compared approaches. In the complex botanical texture case shown in Fig. 4 (f), our method achieves superior fidelity, while competing methods suffer from noticeable blurring effects. The visual results demonstrate the superiority of our method across various challenging in plant image.

### 5.4 ABLATION STUDY

**About our SKRM**. We conduct an ablation study by comparing our SKRM with existing efficient matching methods, including Match & Extraction Module (MEM) Lu et al. (2021) and Coarse-to-Fine Embedded PatchMatch (CFE-PatchMatch) Xia et al. (2022). We calculate GFLOPs for an LR image ($400 \times 199$) and Ref image ($1200 \times 796$). As listed in Tab. 2, our SKRM achieves the lowest computational cost (77.86 GFLOPs) while maintaining the best performance. Compared to the exhaustive Enumerated Matching method, SKRM significantly reduces the computation by over $150 \times$. These results (More results in Appendix G) demonstrate the efficiency of our SKRM.

**About our TGDM**. We conduct an ablation study by comparing our proposed TGDM with existing texture fusion methods, including Dynamic Aggregation (DA) Jiang et al. (2021) and Residual Feature Aggregation (RFA) Cao et al. (2022). As listed in Tab. 3, TGDM achieves the highest performance, outperforming both DA and RFA. To further assess the contribution of the diffusion mechanism in TGDM, we also compare it with a variant that removes the diffusion module. The result shows a performance drop from 38.62 to 38.52 in PSNR, highlighting the importance of the diffusion design. These results validate the effectiveness of our TGDM.

Table 1: Quantitative comparisons of RefSR methods, each independently trained on the CUFED5 Zhang et al. (2019), DRefSR Zhou et al. (2025), and our PlantRSR dataset, with evaluation performed on PlantRSR testing sets. **Bold text** indicates the best results achieved on the respective training datasets.

| Method | Param. | Training set | PSNR↑ | M-PSRN↑ | SSIM↑ | LPIPS↓ | DISTS↓ |
|---|---|---|---|---|---|---|---|
| MASA | 4.0M | CUFED5 | 38.00 | 30.46 | 0.9491 | 0.1344 | 0.0863 |
|  |  | DRefSR | 38.06 | 30.54 | 0.9495 | 0.1338 | 0.0859 |
|  |  | PlantRSR | 38.24 | 30.80 | 0.9510 | 0.1323 | 0.0852 |
| $C^2$-Matching | 8.9M | CUFED5 | 38.23 | 30.81 | 0.9508 | 0.1327 | 0.0857 |
|  |  | DRefSR | 38.27 | 30.87 | 0.9509 | 0.1323 | 0.0855 |
|  |  | PlantRSR | 38.43 | 31.20 | 0.9523 | 0.1300 | 0.0851 |
| AMSA | 9.7M | CUFED5 | 38.24 | 30.83 | 0.9509 | 0.1324 | 0.0851 |
|  |  | DRefSR | 38.29 | 30.89 | 0.9511 | 0.1321 | 0.0849 |
|  |  | PlantRSR | 38.43 | 31.23 | 0.9526 | 0.1303 | 0.0842 |
| DATSR | 18.0M | CUFED5 | 38.25 | 30.81 | 0.9512 | 0.1316 | 0.0842 |
|  |  | DRefSR | 38.32 | 30.91 | 0.9515 | 0.1312 | 0.0839 |
|  |  | PlantRSR | 38.48 | 31.26 | 0.9527 | 0.1304 | 0.0835 |
| RRSR | 21.5M | CUFED5 | 38.26 | 30.83 | 0.9510 | 0.1307 | 0.0838 |
|  |  | DRefSR | 38.31 | 30.93 | 0.9513 | 0.1303 | 0.0835 |
|  |  | PlantRSR | 38.44 | 31.22 | 0.9524 | 0.1299 | 0.0829 |
| MRefSR | 23.7M | CUFED5 | 38.21 | 30.76 | 0.9502 | 0.1334 | 0.0862 |
|  |  | DRefSR | 38.30 | 30.91 | 0.9511 | 0.1328 | 0.0859 |
|  |  | PlantRSR | 38.42 | 31.15 | 0.9516 | 0.1317 | 0.0853 |
| HiTSR | 13.7M | CUFED5 | 37.84 | 30.12 | 0.9482 | 0.1399 | 0.0911 |
|  |  | DRefSR | 37.90 | 30.21 | 0.9487 | 0.1390 | 0.0907 |
|  |  | PlantRSR | 38.07 | 30.49 | 0.9499 | 0.1371 | 0.0900 |
| MCMSR | 8.9M | CUFED5 | 38.26 | 30.82 | 0.9510 | 0.1322 | 0.0851 |
|  |  | DRefSR | 38.30 | 30.90 | 0.9516 | 0.1317 | 0.0849 |
|  |  | PlantRSR | 38.43 | 31.23 | 0.9526 | 0.1302 | 0.0844 |
| SSMTF | 13.9M | CUFED5 | 38.28 | 30.86 | 0.9514 | 0.1305 | 0.0834 |
|  |  | DRefSR | 38.31 | 30.92 | 0.9517 | 0.1303 | 0.0833 |
|  |  | PlantRSR | 38.49 | 31.25 | 0.9528 | 0.1297 | 0.0831 |
| **Ours** | 11.1M | CUFED5 | **38.40** | **31.14** | **0.9522** | **0.1294** | **0.0828** |
|  |  | DRefSR | **38.49** | **31.27** | **0.9529** | **0.1292** | **0.0828** |
|  |  | PlantRSR | **38.62** | **31.53** | **0.9538** | **0.1288** | **0.0826** |

Table 2: Ablation study of our SKRM.

| Matching Method | GFLOPs | PSNR/SSIM |
|---|---|---|
| MEM | 620.72 | 38.36/0.9517 |
| CFE-PatchMatch | 124.73 | 38.58/0.9534 |
| Enumerated Matching | 11990.39 | **38.62/0.9542** |
| SKRM | 77.86 | **38.62**/0.9538 |

Table 3: Ablation study of our TGDM.

| Fusion Method | PSNR/SSIM/M-PSNR |
|---|---|
| DA | 38.43/0.9513/31.18 |
| RFA | 38.48/0.9521/31.25 |
| TGDM (w/o diffusion) | 38.52/0.9531/31.33 |
| TGDM | **38.62/0.9538/31.53** |

**About our PlantRSR dataset**. To validate the effectiveness of our PlantRSR dataset, we train our method on CUFED5, DRefSR, and PlantRSR datasets, respectively, and evaluate the models on PlantRSR testing dataset. As shown in Fig. 5, our model trained on the PlantRSR dataset consistently outperformed the models trained on CUFED5 and DRefSR across the entire training process, with the highest PSNR of 38.62 dB compared to 38.40 dB and 38.50 dB. This quantitative improvement demonstrates that our PlantRSR dataset significantly outperforms existing CUFED5 and DRefSR datasets when handling complex botanical textures. Besides, as shown in Fig. 6 (a), the model trained on the PlantRSR dataset better leverages Ref image to recover fine textures, such as plant trichomes. Fig. 6 (b) exhibits that our dataset achieves superior recovery of irregular floral textures. These results substantiate the effectiveness of our PlantRSR dataset.

**About comparisons of other testing datasets**. To further demonstrate the effectiveness of our method, we conduct comparative experiments on the CUFED5 and WR-SR testing datasets commonly compared in RefSR task (See Appendix D for other datasets). All methods are trained on CUFED5 datasets. As listed in Tab. 4, our method consistently outperforms all competitors across both datasets, achieving superior results (See Appendix E for visual comparison).

**About sampling steps**. We investigate the impact of sampling steps $T$ in our method. As shown in Fig. 7, increasing the number of steps improves the performance, with PSNR gradually saturating after $T = 4$. We also evaluate different texture types by grouping the test images according to their

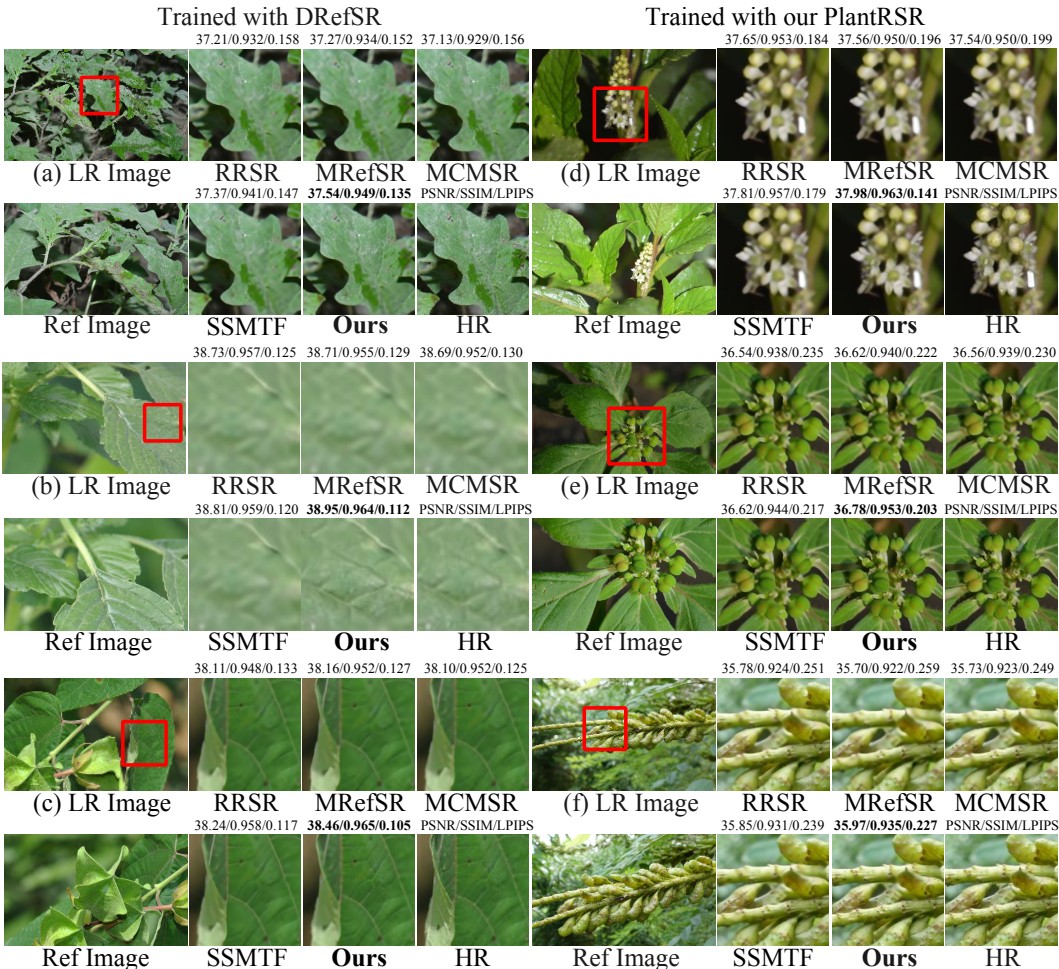

Figure 4: Visual comparison of SOTA RefSR methods, each independently trained on both the DRefSR dataset and our proposed PlantRSR dataset, with evaluation performed on the PlantRSR.

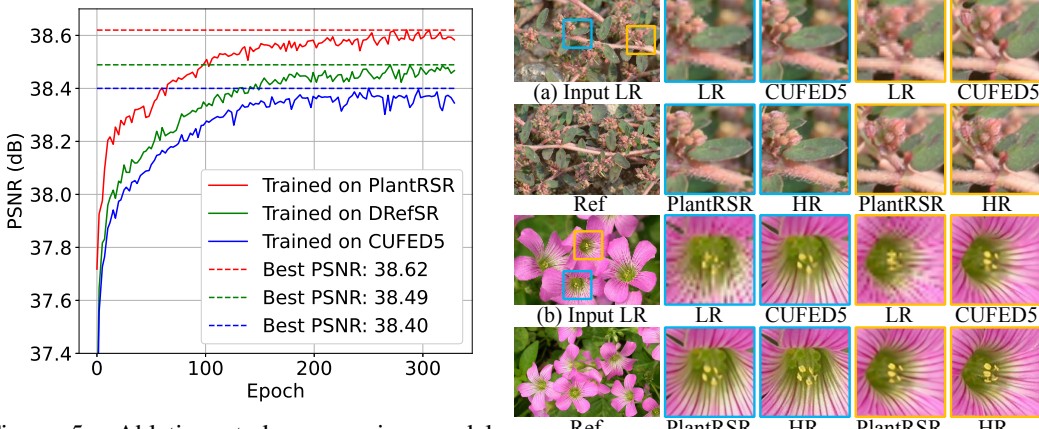

Figure 5: Ablation study comparing model performance when trained on the CUFED5, DRefSR, and our PlantRSR datasets.

Figure 6: Visual comparison between models trained on CUFED5 and our PlantRSR dataset.

complexity, such as fine veins (25 images) versus thick stems (37 images). As shown in Tab. 5, across all groups, we observed the same trend as in Fig. 7: performance steadily improves when increasing $T$ and saturates at $T = 4$. Importantly, no texture category benefits from larger sampling

Table 4: Quantitative comparisons of SOTA methods on CUFED5 and WR-SR testing datasets, evaluated using PSNR, SSIM, and LPIPS.

| Method | CUFED5 | WR-SR |
|--------|--------|-------|
| RRSR | 28.83/0.8563/0.2241 | 28.41/0.8039/0.2846 |
| MRefSR | 28.63/0.8523/0.2281 | 28.26/0.8012/0.2912 |
| HiTSR | 27.08/0.8012/0.2960 | 28.26/0.8017/0.2958 |
| MCMSR | 28.54/0.8490/0.2294 | 28.34/0.8019/0.2857 |
| SSMTF | 28.86/0.8595/0.2202 | 28.42/0.8056/0.2842 |
| **Ours** | **28.95/0.8602/0.2184** | **28.51/0.8062/0.2813** |

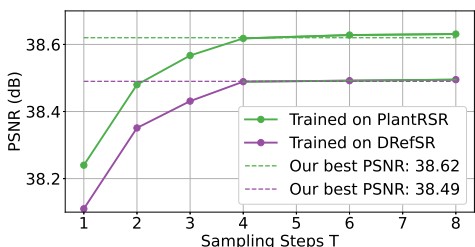

Figure 7: Ablation study of T in TGDM.

steps beyond $T = 4$, confirming that the diffusion refinement behaves consistently across different plant textures. Based on the trade-off between performance and efficiency, we set $T = 4$.

Table 5: About the effects of different sampling steps on different plant textures.

| Texture Type | T=1 | T=2 | T=3 | T=4 | T=6 | T=8 |
|--------------|-----|-----|-----|-----|-----|-----|
| Fine Veins | 27.62 | 27.88 | 27.95 | 28.04 | 28.06 | 28.07 |
| Thick Stems | 36.82 | 37.07 | 37.26 | 37.32 | 37.33 | 37.34 |

**About plant categories.** We provide a clearer description of plant categories, growth-stage diversity, and real-scene complexity. The dataset covers four major categories of commonly encountered vegetation: crops, wild plants, ornamental plants, and aquatic plants. To improve generalization, data acquisition was conducted over more than one year, covering spring, summer, and autumn, thus naturally capturing early leaf expansion, mid-growth, and mature stages. The overall distribution is listed in the Tab. 6

Table 6: About category of our PlantRSR.

| Category | Percentage | Growth-Stage |
|----------|-----------|--------------|
| Crops | 10.2% | early, mid, mature |
| Wild plants | 36.3% | early, mid, mature |
| Ornamental Plants | 45.3% | early, mid, mature |
| Aquatic Plants | 8.2% | mid, mature |

Table 7: About environment of our PlantRSR.

| Environmental Factor | Percentage |
|----------------------|-----------|
| Illumination variations | 22.7% |
| Weather effects | 9.8% |
| Background clutter | 34.6% |
| Leaf damage | 7.3% |

**About collection environment.** The dataset was deliberately constructed to encompass a wide spectrum of natural environmental changes. Images were captured across different seasons to include diverse factors such as illumination variations, weather effects, background clutter, and instances of leaf damage. A quantitative summary of these conditions and their approximate proportions within the dataset is provided in Tab. 7.

## 6 CONCLUSION

In this paper, we address the limitations of existing RefSR datasets and methods in handling natural scenes with complex textures, particularly plant imagery. We introduce **PlantRSR**, a large-scale RefSR dataset containing 16,585 high-quality HR–Ref pairs that capture the diverse and fine-grained characteristics of botanical scenes. To fully exploit this dataset, we propose a novel RefSR framework featuring two key components: a SKRM selectively matches botanical textures to enhance efficiency, and a TGDM for progressive refinement of LR features using Ref-guided diffusion. Extensive experiments demonstrate that our method outperforms existing SOTA approaches in both quantitative metrics and visual quality, highlighting the effectiveness of our design and the value of the PlantRSR dataset for advancing RefSR research in natural and fine-grained scenarios.

## 7 ACKNOWLEDGEMENTS

The research is supported by National Science and Technology Major Project (2022ZD0119204), National Science Fund for Distinguished Young Scholars (62125601), and National Natural Science Foundation of China (62576031).

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

## A    THE USE OF LARGE LANGUAGE MODELS

We used Large Language Models to assist or polish the writing, without involving our experiments, figures, or other core contributions.

## B    ARCHITECTURE OF THE DENOISING NETWORK (DN)

As shown in Fig. 8, DN is designed to restore the latent feature $Z_0$ by leveraging both Ref texture features and diffusion timestep. Given the diffusion timestep $t$, a Embedder maps it into an embedding $t_{\text{emb}} \in \mathbb{R}^d$. This embedding is then fused with the matched Ref features $F_{\text{matched}}^{Ref}$ via a simple addition, and passed through a SiLU activation followed by a linear transformation to yield three modulation vectors $a, b, c \in \mathbb{R}^d$:

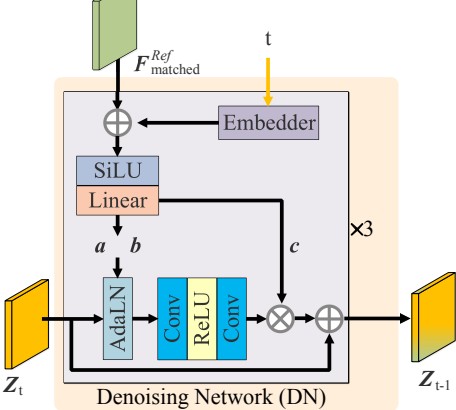

$$t_{\text{emb}} = \text{Embedder}(t), \tag{14}$$

$$a, b, c = \text{Linear}(\text{SiLU}(F_{\text{matched}}^{Ref} + t_{\text{emb}})). \tag{15}$$

The input latent $Z_t$ is passed by the adaptive layer normalization (AdaLN) Perez et al. (2018); Li et al. (2024b). The result is adaptively modulated using the scale vector $a$ and shift vector $b$:

Figure 8: Architecture of Denoising Net.

$$x_t = a \otimes \text{AdaLN}(Z_t) + b. \tag{16}$$

This modulated feature $x_t$ is further refined by a ResBlock He et al. (2016), whose output is scaled by $c$ and added back to the original input $Z_t$ to obtain the $Z_{t-1}$:

$$Z_{t-1} = c \otimes ResBlock(x_t) + Z_t. \tag{17}$$

To enhance the representational capacity and further inject Ref information into the denoising process, the above procedure is repeated three times within the DN. This recursive structure enables more effective integration of Ref textures and temporal priors for progressive feature refinement.

## C COMPARISON WITH THE DRefSR DATASET

Although DRefSR contains some plant scenes, its limited scale and diversity are insufficient to address the challenges in plant imagery. As shown in Tab. 8, DRefSR comprises only 1,400 images, whereas our PlantRSR dataset offers 6,134 high-resolution images, with 80% exceeding 4K resolution and exhibiting superior visual quality. Furthermore, PlantRSR provides more comprehensive coverage across five key categories, while DRefSR contains merely a few dozen images in critical dimensions such as color variation, scale diversity, and background complexity—far below the requirements for robust model training. The experimental results in Tab. 1 and Fig 5 further demonstrate the superior effectiveness of our PlantRSR dataset.

Table 8: Analysis of plant image diversity and number in DRefSR and our PlantRSR.

| Category | Color | Scale | Rotation | Deformation | Background | All |
|---|---|---|---|---|---|---|
| DRefSR | 52 (3.7%) | 83 (5.9%) | 332 (23.7%) | 884 (63.2%) | 49 (3.5%) | 1,400 |
| PlantRSR | 697 (11.4%) | 1,142 (18.6%) | 1,657 (27.0%) | 1,872 (30.5%) | 766 (12.5%) | 6,134 |

## D COMPARISON WITH SOTA METHODS ON OTHER TESTING DATASETS

To further demonstrate the effectiveness of our method, we conduct comparative experiments on the Sun80 Sun & Hays (2012) and DRefSR Zhou et al. (2025) testing datasets commonly compared in RefSR task. For fair comparison, all methods are trained on the CUFED5 dataset. As presented in Tab. 9, our method surpasses all competing approaches across all evaluation metrics on both datasets, demonstrating its consistent superiority and effectiveness.

Table 9: Quantitative comparisons of SOTA methods on Sun80 Sun & Hays (2012) and DRefSR Zhou et al. (2025) testing datasets, evaluated using PSNR, SSIM, and LPIPS.

| Method | Sun80 | DRefSR |
|---|---|---|
| RRSR | 30.13/0.816/0.303 | 31.69/0.867/0.280 |
| MRefSR | 30.28/0.819/0.301 | 31.72/0.868/0.279 |
| HiTSR | 30.24/0.821/0.293 | 31.26/0.858/0.290 |
| MCMSR | 30.21/0.818/0.300 | 31.24/0.856/0.294 |
| SSMTF | 30.38/**0.824**/0.286 | 31.75/0.869/0.277 |
| **Ours** | **30.41/0.824/0.284** | **31.77/0.871/0.272** |

## E VISUAL COMPARISON ON OTHER TESTING DATASETS

To further demonstrate the effectiveness of our method, we conduct comparative experiments on the CUFED5 Zhang et al. (2019) and WR-SR Jiang et al. (2021) testing datasets commonly compared in RefSR task. All methods are trained on CUFED5 datasets. As shown in Fig. 11, our method demonstrates superior visual reconstruction performance compared to other approaches across both datasets.

Table 10: Performance in terms of different similarity levels on CUFED5 dataset. All methods trained on CUFED5 dataset.

| Similarity Levels | DATSR | RRSR | HiTSR | MCMSR | SSMTF | Ours |
|---|---|---|---|---|---|---|
| L1 | 28.50/0.850 | 28.63/0.851 | 26.82/0.797 | 28.54/0.849 | 28.76/0.854 | **28.86/0.855** |
| L2 | 27.47/0.820 | 27.67/0.821 | 26.68/0.785 | 27.54/0.808 | 27.71/0.824 | **27.81/0.826** |
| L3 | 27.22/0.811 | 27.41/0.813 | 26.56/0.783 | 27.27/0.810 | 27.46/0.816 | **27.58/0.819** |
| L4 | 26.96/0.803 | 27.15/0.804 | 26.43/0.781 | 27.03/0.801 | 27.19/0.807 | **27.29/0.809** |
| LR | 25.75/0.754 | 26.53/0.784 | 26.53/0.782 | 26.41/0.782 | 26.68/0.791 | **26.70/0.791** |
| Average | 27.18/0.808 | 27.47/0.815 | 26.60/0.786 | 27.36/0.810 | 27.56/0.818 | **27.65/0.820** |

## F ABOUT SIMILARITY OF REF IMAGE

To evaluate the impact of reference images with varying similarity levels on model performance, which provides four Ref images (L1–L4) with decreasing similarity to the LR image, where L1 is

the most similar and L4 the least. We evaluate our method using all similarity levels, including the LR image itself as a Ref when similar ones are unavailable. As listed in Tab. 10, our method consistently outperforms SOTA approaches across all levels, even when using only the LR image as a Ref, demonstrating its robustness and generalization ability.

Table 11: Computational cost and memory usage of different methods.

| Method | Param. | Runtime | Memory |
|--------|--------|---------|--------|
| RRSR   | 21.5M  | 1.496s  | 13.3G  |
| MRefSR | 23.7M  | 0.774s  | 14.1G  |
| HiTSR  | 13.7M  | 0.843s  | 8.5G   |
| MCMSR  | 8.9M   | 0.681s  | 8.6G   |
| SSMTF  | 13.9M  | 1.435s  | 15.2G  |
| Ours   | 11.1M  | 1.116s  | 11.4G  |

## G COMPUTATIONAL COST AND MEMORY USAGE

To assess the practicality of our method, we compare the runtime and memory consumption with several recent RefSR methods in Tab. 11. All values are measured using an LR image of size $300 \times 200$ and a Ref image of size $1200 \times 800$. As listed in Tab. 11, while our method does not achieve the lowest runtime or memory usage, it maintains a reasonable balance between efficiency and performance. Although we adopt a diffusion process, the network structure is lightweight, consisting of only three ResBlocks, and the sampling steps during inference are limited to four. Therefore, the computational and time costs are not as significant. In addition, our SKRM module effectively reduces the time required for Ref texture matching, resulting in competitive overall runtime and memory usage.

Table 12: Comparison with diffusion-based methods on PlantRSR dataset.

| Method | PSNR | SSIM | LPIPS | DISTS |
|--------|------|------|-------|-------|
| SinSR Wang et al. (2024b) | 31.60 | 0.8580 | 0.1886 | 0.1449 |
| DoSSR Cui et al. (2024) | 30.82 | 0.8600 | 0.2192 | 0.2091 |
| StableSR Wang et al. (2024a) | 31.21 | 0.8774 | 0.2011 | 0.1930 |
| OSEDiff Wu et al. (2024) | 32.22 | 0.8975 | 0.1602 | 0.1312 |
| Ours | 38.62 | 0.9538 | 0.1288 | 0.0826 |

## H COMPARISON WITH DIFFUSION-BASED METHODS

We conduct comparative experiments with diffusion-based methods, as listed in Tab. 12. It should be noted that this comparison may present certain inequities and is primarily intended for experimental reference. These diffusion-based approaches are built upon generative diffusion models, while our method is trained using pixel-wise losses (e.g., L1 loss). Furthermore, the diffusion architectures are substantially larger and more complex, whereas our method adopts a lightweight design comprising only three ResBlocks. Additionally, these comparative methods do not utilize reference images, which places them at a inherent disadvantage in the reference-based super-resolution setting.

Table 13: The user study. Compared to other methods, over 90% users prefer our results.

| Ours vs. RRSR | Ours vs. MRefSR | Ours vs. MCMSR | Ours vs SSMTF |
|---------------|-----------------|----------------|---------------|
| 93% | 95.3% | 93% | 90.7% |

## I USER STUDY

To further validate the qualitative superiority, as listed in the table below, we conduct a user study involving 43 participants to compare the visual quality of our method against several methods on our PlantRSR dataset, including RRSR, MRefSR, HiTSR, MCMSR, and SSMTF. In each comparison, participants are shown image pairs, with one generated by Ours, and are asked to select the image

with better visual quality. According to the Tab. 13, over 90% of participants favor the results of our method over the other approaches.

## J    LIMITATION

While this work advances RefSR for plant imagery, two key limitations should be noted. First, hardware constraints force suboptimal downsampling of our high-resolution dataset (80% > 4K, range 2K-8K), preventing full utilization of the available image detail. Second, performance degrades when processing dissimilar Ref images, as the current matching mechanism lacks adaptability to low-similarity scenarios. These limitations point to important research directions: developing memory-efficient architectures for ultra-high-resolution processing and designing more selective Ref utilization strategies. Addressing these challenges would significantly enhance the practical applicability of plant image SR.

## K    PARTIAL SAMPLES FROM THE PLANTRSR DATASET

As shown in Fig. 9 and Fig. 10, our PlantRSR dataset contains meticulously annotated samples spanning multiple plant categories with rich textures. The manual annotation process specifically captures challenging plant scenarios, ensuring dataset quality for advanced botanical studies.

## L    ETHICS STATEMENT

Our proposed dataset focuses on plant scene imagery, and our method is designed for reference-based image super-resolution. Neither the dataset nor the proposed method involves human subjects, personal data, or sensitive information. The dataset was constructed from images captured by our team and does not infringe upon the interests of any individuals or communities. All experimental results are derived from either publicly available datasets or our constructed dataset. We do not foresee any direct social harms arising from this research.

## M    REPRODUCIBILITY STATEMENT

Our codes are released at: `https://github.com/edbca/PlantRSR`. The generation process of our PlantRSR dataset is presented in Section 3. The training details of our proposed method are described in Section 5.1 of the main text. More details of denoising network are given in Appendix B, while Appendix K provides more samples from the PlantRSR training and testing datasets. The usage of large language models is discussed in Appendix A.

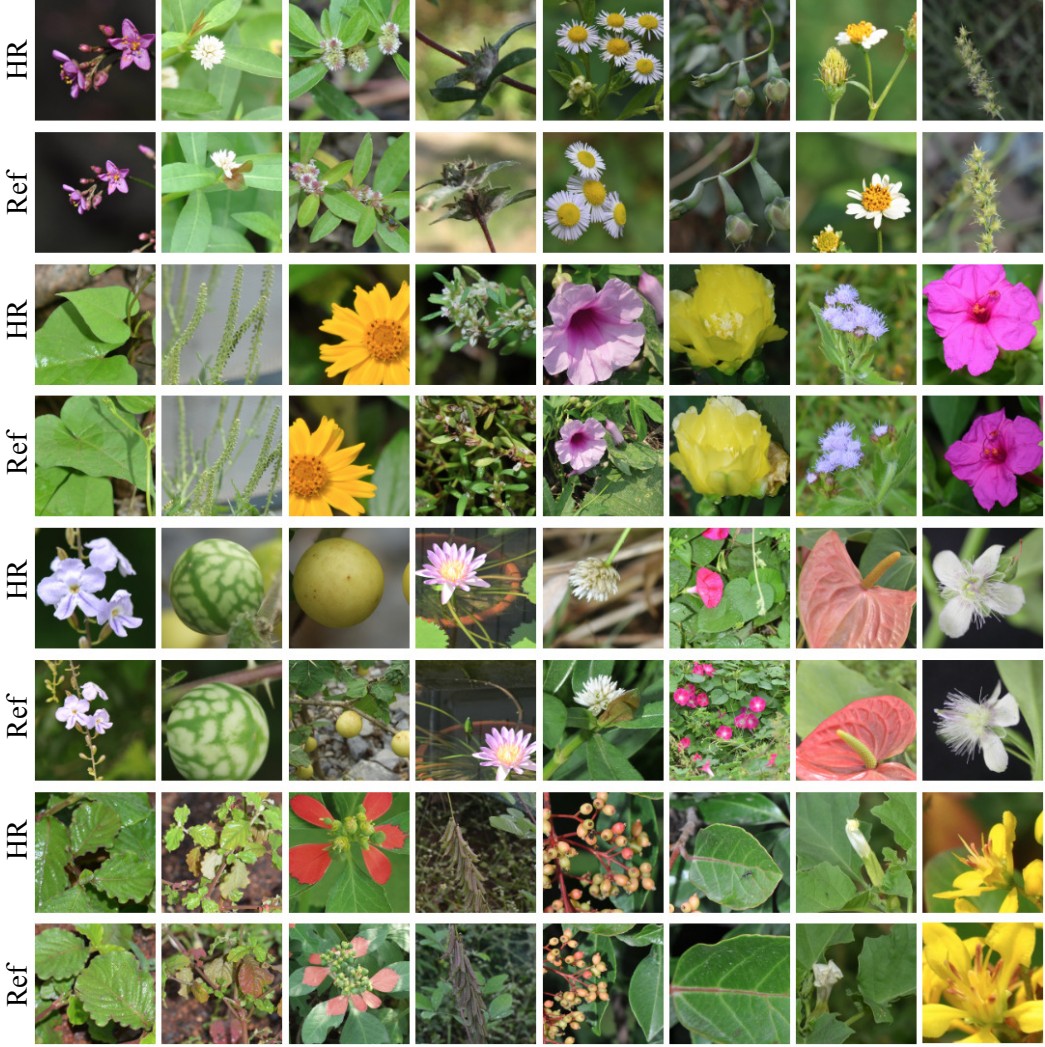

Figure 9: Partial Samples of our PlantRSR training dataset.

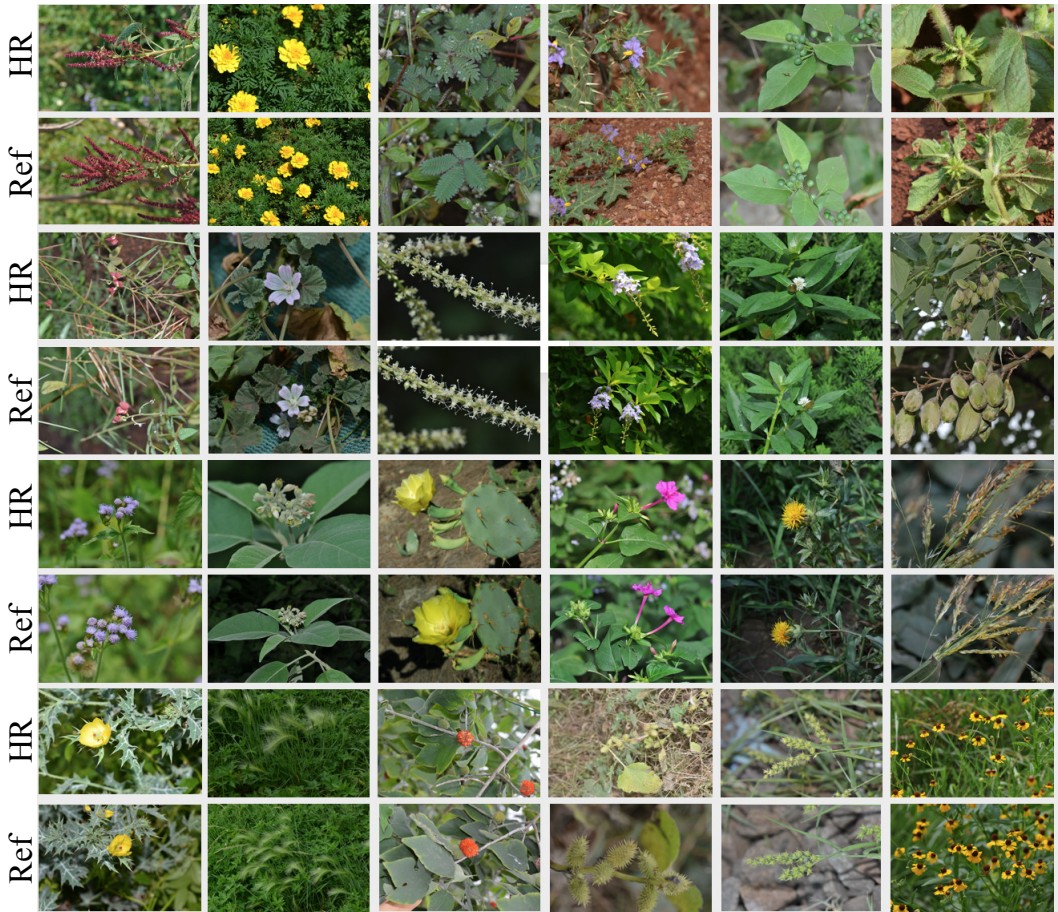

Figure 10: Partial Samples of our PlantRSR testing dataset.

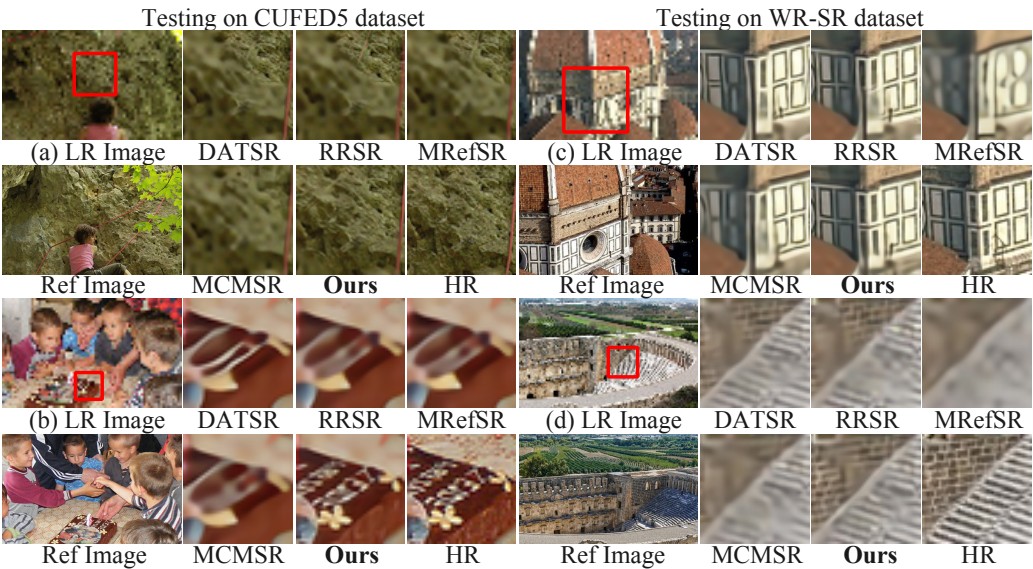

Figure 11: Visual comparison with SOTA methods on CUFED5 and WR-SR datasets.

