# OpenReview forum: "PlantRSR: A New Plant Dataset and Method for Reference-based Super-Resolution"
_ICLR.cc/2026/Conference — ICLR 2026 Poster_

### Official Review · Reviewer_VMqn · 2025-10-19

**Soundness:** 3
**Presentation:** 3
**Contribution:** 3
**Rating:** 6
**Confidence:** 4

**Summary:**

This paper presents a new reference-based super-resolution (RefSR) method for plant imagery. Potential contributions are threefold:
(1) a large-scale PlantSR dataset for plant imagery,
(2) a Selective Key-Region Matching (SKRM) for efficient LR-Ref correspondence,
and (3) a Texture-Guided Diffusion Module (TGDM) that leverages a reference-texture-conditioned diffusion process for feature enhancement.
The method is evaluated on the new PlantRSR and existing general RefSR datasets, demonstrating competitive performance.

**Strengths:**

**Clarity and Reproducibility:**
The paper is generally well-written and easy to follow. The inclusion of the source code strengthens the paper's credibility and reproducibility. However, it is important to release both code and data for full reproducibility.

**New Dataset:**
The efforts for constructing a large-scale dataset are appreciated. This RefSR dataset for plant imagery may enable new research about, e.g., agricultural and forestrial imaging techniques.


**Effective Method Designs:**
SKRM demonstrates balanced performance-efficiency, as evidenced by comparisons in Table 2 and Table 8 (Appendix). TGDM also shows clear advantages over existing fusion methods in Table 3.

**Weaknesses:**

**1. Dataset and Method Significance:**
The significance of introducing a large-scale and domain-specific dataset requires further establishment.

***(1) Downstream Impact:***
Demonstrating downstream applications (e.g., improved plant disease detection accuracy) of plant imagery super-resolution would significantly strengthen the practical value of this work.


***(2) Generalization:***
An important question is whether learning to address the challenges of RefSR in plant imagery leads to models to generalize. The authors are suggested to investigate whether a model trained (or fine-tuned) on this PlantSR dataset can achieve improved performance on existing non-plant RefSR datasets. This helps demonstrate a broader value of PlantSR towards developing more powerful RefSR approaches.


***(3) Dataset Split Rigor:***
The extremely high train-test split ratio (around 60:1) choice requires justification. A discussion on how the 100 test images were selected to represent the dataset's diversity and complexity is necessary for a reliable RefSR evaluation for plant imagery.


**2. Technical Novelty of TGDM and its Evaluation:**
While TGDM is claimed as one of the major contributions, its technical novelty is limited, as a combination of existing techniques, including the conditional diffusion process (Ho et al., 2020; Li et al. 2024), the Residual State Space Block (Gun et al., 2024), and the sub-pixel convolution (She et al. 2016).

***(1) Incomplete Ablation Study:***
The ablation study in Table 3 is incomplete. To evaluate its internal designs, a detailed ablation within the TGDM is required, including the Residual State Space Block and the sub-pixel convolution. Without this, the module feels like a black box where its true novelty remains unclear.


***(2) Marginal Gains vs. Computational Cost:***
The performance gain from the diffusion step appears marginal (e.g., +0.1PSNR and +0.0007SSIM). Given the well-known computational overhead of diffusion models, a clear analysis of the performance-efficiency trade-off is necessary to justify this choice. What is the additional inference time/FLOPs? Is there any significant visual improvement?


***(3) Unfair Comparisons to Diffusion Baselines***
The comparisons with diffusion-based methods in Table 9 (Appendix) are unfair, as those methods are not designed or trained for the RefSR task. A fair evaluation would require modifying and training these baselines with the same RefSR framework (using the reference images and the same losses). Without this, the evaluation on the diffusion step of TGDM is not informative.


**3. In-Depth Discussion on Limitation:**
The authors acknowledge performance degradations in processing dissimilar LR-Ref pairs, which is appreciated. To strengthen the depth of discussion, the authors are suggested to answer:
(1) To what extent of dissimilarity would the method fail? Would it be possible to use semantic feature distance to measure the dissimilarity and plot performance (PSNR/SSIM) against this metric? This may provide concrete insight into the operational boundaries of their method.

(2) To what extent of degradation would the method suffer when processing unmatched LR-Ref image pairs? Considering diffusion is used in this method, would it generate new and unwanted patterns?

**Justification for Recommendation** This paper presents a new dataset and a new RefSR method, with promising results. Meanwhile, the broader impact of the dataset and the novelty/necessity of TGDM require a more rigorous establishment. The core issue lies in reframing the contribution from "a good method for plant image RefSR" to "a dataset and method that advances the general understanding and capability of RefSR".

**Questions:**

**(1) Dataset Impact:** Beyond a new benchmark, can the authors demonstrate the practical impact of the proposed PlantSR dataset? For instance, does using the proposed RefSR method on this data lead to improved performance on a downstream task like plant species classification or disease detection?

**(2) Generalization:** Does training on this plant-specific dataset produce any generalizable knowledge? Please consider reporting results of a model (pre-trained or fine-tuned on PlantSR) evaluated on a standard, non-plant RefSR benchmark (e.g., CUFED5).

**(3) TGDM Novelty & Ablation:** Please provide an internal ablation study (e.g., removing the RSSB, and modifying the conditioning mechanism). Can the authors provide a runtime/FLOPs analysis to justify the cost-to-benefit ratio of the diffusion step inside TGDM?

**(4) Limitations:** Can the authors provide a more quantitative analysis of this method's limitations?

---

> ### Author Response · Authors · 2025-11-20
> **Rebuttals (1/6)**
>
> We sincerely appreciate the time and effort you have devoted to reviewing our manuscript.
>
> > **Weakness 1.1: Downstream Impact: Demonstrating downstream applications (e.g., improved plant disease detection accuracy) of plant imagery super-resolution would significantly strengthen the practical value of this work.**
>
> **Response:** Thanks for this insightful question. To demonstrate the practical utility of the PlantSR dataset beyond benchmarking, we performed a downstream evaluation on plant species classification using three classifiers with different network structures. We applied our model as a pre-processing step on PlantCLEF2022 images, without retraining the classifiers. Since our method is reference-based, each image itself was used as its own reference. This workflow allows fully automatic enhancement of the images before classification. The results are listed in the table below :
>
> |Classifier|Input|MA-MRR|
> |---------- | ------------ | --------- |
> |Model1 [1]|Original| 0.637|
> |         |RefSR (Ours)|0.653 (+0.016)|
> | Model2 [2]|Original| 0.553|
> |         |RefSR (Ours)|0.579 (+0.026)|
> | Model3 [3]|Original|0.460|
> |            |RefSR (Ours)|0.492 (+0.032)|
>
> This procedure shows that SR models trained on PlantSR can improve image quality in a way that benefits downstream plant classification, demonstrating the practical impact of the dataset beyond serving as a benchmark.
>
> Reference:
>
> [1] Xu M., et al., PlantCLEF2023: A bigger training dataset contributes more than advanced pretraining methods for plant identification. In: CLEF, 2023.
> [2] Chulif S., et al., A global-scale plant identification using deep learning: NEUON submission to PlantCLEF 2022. In: CLEF, 2022.
> [3] Carranza R., et al., Extreme automatic plant identification under constrained resources. In: CLEF, 2022.
>
> > **Weakness 1.2: An important question is whether learning to address the challenges of RefSR in plant imagery leads to models to generalize. The authors are suggested to investigate whether a model trained (or fine-tuned) on this PlantSR dataset can achieve improved performance on existing non-plant RefSR datasets. This helps demonstrate a broader value of PlantSR towards developing more powerful RefSR approaches.**
>
> **Response:** Thanks for the valuable suggestion. To evaluate whether training on our plant-specific dataset provides transferable knowledge, we conducted additional experiments following your recommendation. Specifically, for three representative RefSR methods (MCMSR, SSMTF, and ours), we first trained each model on the CUFED5 dataset and then fine-tuned it on our PlantRSR dataset. We subsequently evaluated all models on four standard RefSR benchmarks: CUFED5, WR-SR, Sun80, and DRefSR. As listed in the table below, fine-tuning on PlantRSR consistently improves performance across all datasets, indicating that the plant-focused supervision indeed contributes generalizable texture-alignment and reconstruction capability. The gains on CUFED5 are relatively smaller because its test set is dominated by human-centric scenes with limited plant content, aligning with our analysis in Fig. 1 (a). In contrast, the other datasets contain outdoor environments or dedicated plant imagery, allowing PlantRSR-derived knowledge to transfer more effectively and resulting in larger performance improvements.
>
> | Method | CUFED5 | WR-SR | Sun80 | DRefSR |
> |-------------------|--------|--------|--------|--------|
> |MCMSR (trained on CUFED5)| 28.54 | 28.34 | 30.21 | 31.24 |
> |MCMSR (fine-tuned on PlantRSR) | 28.61(+0.07) | 28.43(+0.09) | 30.31(+0.10) | 31.36(+0.12) |
> | SSMTF (trained on CUFED5) | 28.86 | 28.42 | 30.38 | 31.75 |
> | SSMTF (fine-tuned on PlantRSR) | 28.92(+0.06) | 28.52(+0.10) | 30.49(+0.11) | 31.86(+0.10) |
> | **Ours** (trained on CUFED5) | 28.95 | 28.51 | 30.41 | 31.77 |
> | **Ours** (fine-tuned on PlantRSR) |29.01(+0.06)|28.62(+0.11)|30.53(+0.12)|31.90(+0.13)|
>
> >**Weakness 1.3: The extremely high train-test split ratio (around 60:1) choice requires justification. A discussion on how the 100 test images were selected to represent the dataset's diversity and complexity is necessary for a reliable RefSR evaluation for plant imagery.**
>
> **Response:** Thanks for pointing out this important issue. Although our dataset contains a large number of high-resolution plant images, the selection of the 100-image test set was conducted with strict control to ensure fair and representative RefSR evaluation. Specifically, as shown in the table below, the test set preserves the natural distribution of the five key variation types (color, scale, rotation, deformation, and background changes) closely matching their proportions in the full dataset. This ensures that the test set reflects the same diversity and complexity encountered in real-world plant imagery.
> |PlantRSR|Color|Scale|Rotation|Deformation|Background|
> |-------|-------|-------|-------|-------|-------|
> | 100 (100%) |13 (13%)|20 (20%)|26 (26%)|30 (30%)|11 (11%)|

---

> ### Author Response · Authors · 2025-11-20
> **Rebuttals (2/6)**
>
> > **Weakness 2.1: The ablation study in Table 3 is incomplete. To evaluate its internal designs, a detailed ablation within the TGDM is required, including the Residual State Space Block and the sub-pixel convolution. Without this, the module feels like a black box where its true novelty remains unclear.**
>
> **Response :** Thanks for your constructive suggestion. We conducted additional experiments focusing on the Residual State Space Block (RRSB) and the sub-pixel convolution. Since the sub-pixel convolution is responsible for feature upscaling, removing it entirely prevents the model from producing valid SR outputs. Therefore, for a fair ablation, we replace the sub-pixel layer with a Deconvolution (Transposed Convolution) layer with matched upsampling capability. The results are shown below:
>
> |RRSB| sub-pixel |Deconv|PSNR/SSIM|
> |---------|---------|---------|--------|
> |  | ✓ |  |38.54/0.9521|
> | ✓ |  | ✓ | 38.62/0.9536 |
> | ✓ | ✓ |  | 38.62/0.9538 |
>
> When RRSB is removed, performance drops noticeably (−0.08 dB), confirming its importance in modeling long-range dependencies and fusing texture cues effectively. While sub-pixel convolution and deconvolution achieve comparable reconstruction performance in our setting, sub-pixel convolution offers a more lightweight and implementation-friendly upsampling strategy because it introduces no additional learnable parameters and avoids the instability sometimes observed in deconvolution-based upsampling. This makes sub-pixel convolution a more suitable choice for our TGDM design. The best performance is achieved when RRSB and sub-pixel convolution are used together, demonstrating that TGDM’s internal design is not a black box but a combination of two synergistic components.
>
> > **Weakness 2.2: The performance gain from the diffusion step appears marginal (e.g., +0.1PSNR and +0.0007SSIM). Given the well-known computational overhead of diffusion models, a clear analysis of the performance-efficiency trade-off is necessary to justify this choice. What is the additional inference time/FLOPs?**
>
> **Response:** Thanks for raising this concern. The seemingly small PSNR/SSIM gain mainly results from the characteristics of plant imagery: during data collection, we intentionally captured images with clean backgrounds while focusing on plant textures, leading to situations where the discriminative high-frequency structures occupy only a small portion of the full image. As a result, global metrics tend to dilute improvements made specifically in textured regions. To better reflect the actual enhancement brought by the diffusion step, we additionally report M-PSNR, a masked version of PSNR that evaluates only the texture areas defined by Eq. (1) in our paper. As shown in the table, while the overall PSNR improves modestly, the gain in M-PSNR is substantially larger (e.g., +0.72 dB from T=1 to T=4), indicating that diffusion meaningfully enhances fine-grained plant details such as leaf edges and venation patterns. Moreover, we provide a performance-efficiency analysis: although increasing the number of diffusion steps raises the GFLOPs, the reconstruction quality saturates after T = 4, which we adopt as the final setting to balance accuracy and computational cost.
>
> | T | GFLOPs |M-PSNR | PSNR |
> |---------|---------|---------|--------|
> | T=1 | 1855.53 | 30.81 | 38.24 |
> | T=2 | 2344.26 | 31.22 | 38.48 |
> | T=3 | 2833.71 | 31.35 | 38.57 |
> | T=4 | 3322.31 | 31.53 | 38.62 |
> | T=5 | 3811.48 | 31.55 | 38.63 |
> | T=6 | 4300.27 | 31.56 | 38.64|

---

> ### Author Response · Authors · 2025-11-20
> **Rebuttals (3/6)**
>
> > **Weakness 2.3: Unfair Comparisons to Diffusion Baselines The comparisons with diffusion-based methods in Table 9 (Appendix) are unfair, as those methods are not designed or trained for the RefSR task. A fair evaluation would require modifying and training these baselines with the same RefSR framework (using the reference images and the same losses). Without this, the evaluation on the diffusion step of TGDM is not informative.**
>
> **Response:** Thanks for pointing out this issue. We agree that the diffusion-based baselines in Table 9 were not originally designed or trained for the RefSR task, and therefore the comparison is inherently unfair. As clarified in our paper, our intention was not to claim superiority over these methods but simply to provide reference results, since these diffusion models aim at high-quality image generation rather than Ref-guided reconstruction. In response to your suggestion, we conducted additional experiments to ensure a more equitable comparison: we incorporated the Ref image as an auxiliary input and retrained all diffusion baselines using the same supervision losses and training settings as our method. It is worth noting that several of these models are not structurally compatible with RefSR, meaning that the Ref image cannot be fully utilized and identical losses may not be optimal for them. Nevertheless, we made our best effort to adapt each method fairly. The updated results are shown below:
>
> | Method | PSNR | SSIM | LPIPS | DISTS |
> |---------|---------|---------|--------|--------|
> |SinSR | 32.75 | 0.8441 | 0.1877 |0.1310 |
> |DoSSR | 30.36 | 0.8612 | 0.1981 | 0.1997 |
> |StableSR | 31.79 | 0.8540 | 0.1919 | 0.1893 |
> |OSEDiff | 32.31 | 0.8824 | 0.1766 | 0.1293 |
> |Ours | 38.62 | 0.9538 | 0.1288 | 0.0826 |
>
> These results confirm that even under aligned training settings, conventional diffusion frameworks still struggle to leverage the Ref image effectively for structural recovery, whereas our method is explicitly designed for RefSR and thus achieves significantly better reconstruction performance.
>
> > **Weakness 3.1: To what extent of dissimilarity would the method fail? Would it be possible to use semantic feature distance to measure the dissimilarity and plot performance (PSNR/SSIM) against this metric? This may provide concrete insight into the operational boundaries of their method.**
>
> **Response :** Thanks for this constructive suggestion. To analyze the operational boundary of our method, we conducted experiments on the CUFED5 dataset, which provides four predefined similarity levels (L1–L4) from high to low. These similarity levels are originally determined by SIFT feature matching. Following the reviewer’s suggestion, we further computed semantic similarity using CLIP feature distance to provide a more perceptually aligned metric of cross-image similarity. The results are shown below. Both SIFT-based and CLIP-based similarity scores consistently decrease from L1 to L4, confirming that the four levels represent progressively larger dissimilarities. Correspondingly, our RefSR performance (PSNR/SSIM) also degrades monotonically as the reference becomes less similar, indicating a smooth and predictable failure mode.**
>
> | Similarity Levels | SIFT feature matching | CLIP feature matching | PSNR/SSIM |
> |-------------------|--------|--------|--------|
> | L1 | 0.611 | 0.637 | 28.86/0.855 |
> | L2 | 0.421 | 0.585 | 27.81/0.826 |
> | L3 | 0.313 | 0.433 | 27.58/0.819 |
> | L4 | 0.204 | 0.221 | 27.29/0.809 |

---

> ### Author Response · Authors · 2025-11-20
> **Rebuttals (4/6)**
>
> > **Weakness 3.2: To what extent of degradation would the method suffer when processing unmatched LR-Ref image pairs? Considering diffusion is used in this method, would it generate new and unwanted patterns?**
>
> **Response :** Thanks for the insightful question. The degradation trend under unmatched LR–Ref pairs is quantitatively reflected in Table 7 of our paper and the table below, where CUFED5 provides references with four predefined similarity levels (L1–L4, from high to low). As similarity decreases, our method exhibits a smooth and monotonic performance drop rather than catastrophic failure. Even at the lowest similarity level (L4), the performance reduction is limited (−1.57 dB compared with L1), and our method remains consistently competitive among recent approaches. Regarding the diffusion component, we note that it does not generate hallucinated or unwanted patterns in low-similarity cases. This is because: (1) Diffusion in our framework is reference-guided rather than unconditional. (2) When the reference becomes unreliable, the LR structure acts as a strong constraint, preventing the model from synthesizing non-existent textures. This behavior is also reflected in the quantitative results: when using the LR image itself as the reference (“LR”), the performance naturally decreases but remains stable and free of hallucinated artifacts. Overall, the results confirm that our method degrades gracefully with decreasing similarity without producing unwanted diffusion artifacts.
>
> | Similarity Levels | DATSR | RRSR | HiTSR | MCMSR | SSMTF | Ours |
> |-------------------|--------|--------|--------|--------|--------|--------|
> | L1 | 28.50/0.850 | 28.63/0.851 | 26.82/0.797 | 28.54/0.849 | 28.76/0.854 |**28.86/0.855**|
> | L2 | 27.47/0.820 | 27.67/0.821 | 26.68/0.785 | 27.54/0.808 | 27.71/0.824 |**27.81/0.826**|
> | L3 | 27.22/0.811 | 27.41/0.813 | 26.56/0.783 | 27.27/0.810 | 27.46/0.816 |**27.58/0.819**|
> | L4 | 26.96/0.803 | 27.15/0.804 | 26.43/0.781 | 27.03/0.801 | 27.19/0/807 |**27.29/0.809**|
> | LR | 25.75/0.754 | 26.53/0.784 | 26.53/0.782 | 26.41/0.782 | 26.68/0.791 |**26.70/0.791**|
> |**Average**|27.18/0.808|27.47/0.815|26.60/0.786|27.36/0.810|27.56/0.818|**27.65/0.820**|
>
> > **Question1: Dataset Impact: Beyond a new benchmark, can the authors demonstrate the practical impact of the proposed PlantSR dataset? For instance, does using the proposed RefSR method on this data lead to improved performance on a downstream task like plant species classification or disease detection?**
>
> **Response :** Thanks for this insightful question. To demonstrate the practical utility of the PlantSR dataset beyond benchmarking, we performed a downstream evaluation on plant species classification using three classifiers with different network structures. We applied our model as a pre-processing step on PlantCLEF2022 images, without retraining the classifiers. Since our method is reference-based, each image itself was used as its own reference. This workflow enables the fully automatic enhancement of images prior to classification. The results are listed in the table below:
>
> | Classifier | Input | MA-MRR |
> | ---------- | ------------ | --------- |
> | Model1 [1]    | Original     | 0.637      |
> |            | RefSR (Ours) | 0.653 (+0.016) |
> | Model2 [2]    | Original     | 0.553      |
> |            | RefSR (Ours) | 0.579 (+0.026) |
> | Model3 [3]   | Original     | 0.460      |      |
> |            | RefSR (Ours) | 0.492 (+0.032) |
>
> This procedure shows that SR models trained on PlantSR can improve image quality in a way that benefits downstream plant classification, demonstrating the practical impact of the dataset beyond serving as a benchmark.
>
> Reference:
> [1] Xu M., et al., PlantCLEF2023: A bigger training dataset contributes more than advanced pretraining methods for plant identification. In: CLEF, 2023.
> [2] Chulif S., et al., A global-scale plant identification using deep learning: NEUON submission to PlantCLEF 2022. In: CLEF, 2022.
> [3] Carranza R., et al., Extreme automatic plant identification under constrained resources. In: CLEF, 2022.

---

> ### Author Response · Authors · 2025-11-20
> **Rebuttals (5/6)**
>
> > **Question2: Does training on this plant-specific dataset produce any generalizable knowledge? Please consider reporting results of a model (pre-trained or fine-tuned on PlantSR) evaluated on a standard, non-plant RefSR benchmark (e.g., CUFED5).**
>
> **Response :** Thanks for the valuable suggestion. To evaluate whether training on our plant-specific dataset provides transferable knowledge, we conducted additional experiments following your recommendation. Specifically, for three representative RefSR methods (MCMSR, SSMTF, and ours), we first trained each model on the CUFED5 dataset and then fine-tuned it on our PlantRSR dataset. We subsequently evaluated all models on four standard RefSR benchmarks: CUFED5, WR-SR, Sun80, and DRefSR. As shown in the table below, fine-tuning on PlantRSR consistently improves performance across all datasets, indicating that the plant-focused supervision indeed contributes generalizable texture-alignment and reconstruction capability. The gains on CUFED5 are relatively smaller because its test set is dominated by human-centric scenes with limited plant content, aligning with our analysis in Fig. 1(a). In contrast, the other datasets contain outdoor environments or dedicated plant imagery, allowing PlantRSR-derived knowledge to transfer more effectively and resulting in larger performance improvements.
>
> |Method|CUFED5|WR-SR|Sun80|DRefSR|
> |---|---|---|---|---|
> |MCMSR (trained on CUFED5)|28.54|28.34|30.21|31.24|
> |MCMSR (fine-tuned on PlantRSR) |28.61(+0.07)|28.43(+0.09)|30.31(+0.10)|31.36(+0.12)|
> | SSMTF (trained on CUFED5) |28.86|28.42|30.38|31.75|
> | SSMTF (fine-tuned on PlantRSR) |28.92(+0.06)|28.52(+0.10)|30.49(+0.11)|31.86(+0.10)|
> | **Ours** (trained on CUFED5) |28.95| 28.51|30.41|31.77|
> | **Ours** (fine-tuned on PlantRSR) |29.01(+0.06)|28.62(+0.11)|30.53(+0.12)|31.90(+0.13)|
>
> > **Question3: Please provide an internal ablation study (e.g., removing the RSSB, and modifying the conditioning mechanism). Can the authors provide a runtime/FLOPs analysis to justify the cost-to-benefit ratio of the diffusion step inside TGDM?**
>
> **Response :** Thanks for your constructive suggestion. We conducted additional experiments focusing on the Residual State Space Block (RRSB) and the sub-pixel convolution. Since the sub-pixel convolution is responsible for feature upscaling, removing it entirely prevents the model from producing valid SR outputs. Therefore, for a fair ablation, we replace the sub-pixel layer with a Deconvolution (Transposed Convolution) layer with matched upsampling capability. The results are shown below, when RRSB is removed, performance drops noticeably (−0.08 dB), confirming its importance in modeling long-range dependencies and fusing texture cues effectively. While sub-pixel convolution and deconvolution achieve comparable reconstruction performance in our setting, sub-pixel convolution offers a more lightweight and implementation-friendly upsampling strategy because it introduces no additional learnable parameters and avoids the instability sometimes observed in deconvolution-based upsampling. This makes sub-pixel convolution a more suitable choice for our TGDM design. The best performance is achieved when RRSB and sub-pixel convolution are used together, demonstrating that TGDM’s internal design is not a black box but a combination of two synergistic components.
>
> | RRSB |Sub-pixel| Deconv | PSNR/SSIM |
> |---|----|---|---|
> |  | ✓ |  |38.54/0.9521|
> | ✓ |  | ✓ | 38.62/0.9536 |
> | ✓ | ✓ |  | 38.62/0.9538 |
>
> To further address your concern about the cost-benefit ratio of diffusion, we provide a runtime/FLOPs analysis by varying the number of diffusion steps T. During data collection, we intentionally captured images with clean backgrounds while focusing on plant textures, leading to situations where the discriminative high-frequency structures occupy only a small portion of the full image. As a result, global metrics tend to dilute improvements made specifically in textured regions. To better reflect the actual enhancement brought by the diffusion step, we additionally report M-PSNR, a masked version of PSNR that evaluates only the texture areas defined by Eq. (1) in our paper. As shown in the table, while the overall PSNR improves modestly, the gain in M-PSNR is substantially larger (e.g., +0.72 dB from T=1 to T=4), indicating that diffusion meaningfully enhances fine-grained plant details such as leaf edges and venation patterns. Moreover, we provide a performance-efficiency analysis: although increasing the number of diffusion steps raises the GFLOPs, the reconstruction quality saturates after T = 4, which we adopt as the final setting to balance accuracy and computational cost.
>
> | T | GFLOPs |M-PSNR | PSNR |
> |---|---|---|---|
> | T=1 | 1855.53 | 30.81 | 38.24 |
> | T=2 | 2344.26 | 31.22 | 38.48 |
> | T=3 | 2833.71 | 31.35 | 38.57 |
> | T=4 | 3322.31 | 31.53 | 38.62 |
> | T=5 | 3811.48 | 31.55 | 38.63 |
> | T=6 | 4300.27 | 31.56 | 38.64|

---

> ### Author Response · Authors · 2025-11-20
> **Rebuttals (6/6)**
>
> > **Question4: Limitations: Can the authors provide a more quantitative analysis of this method's limitations?**
>
> **Response :** Thanks for the suggestion. We provide a more quantitative analysis of the method’s limitations by examining performance under varying LR–Ref similarity conditions. Similar to other RefSR approaches, our method inevitably degrades when the similarity between the LR image and its reference decreases, because fewer transferable textures remain available. To quantify this effect, we evaluate our method on the CUFED5 benchmark, which offers four predefined similarity levels (L1–L4, from high to low). We additionally include an extreme case where the LR image itself is used as the reference. As shown in the table below, all methods exhibit a consistent performance drop as the similarity decreases, confirming this inherent limitation of reference-based frameworks. However, across all similarity levels, including the low-similarity setting (L4) and the LR-only condition, our method maintains the highest PSNR and SSIM, demonstrating stronger robustness compared with prior work.
>
> | Similarity Levels | DATSR | RRSR | HiTSR | MCMSR | SSMTF | **Ours** |
> |-------------------|--------|--------|--------|--------|--------|--------|
> | L1 | 28.50/0.850 | 28.63/0.851 | 26.82/0.797 | 28.54/0.849 | 28.76/0.854 | **28.86/0.855** |
> | L2 | 27.47/0.820 | 27.67/0.821 | 26.68/0.785 | 27.54/0.808 | 27.71/0.824 | **27.81/0.826** |
> | L3 | 27.22/0.811 | 27.41/0.813 | 26.56/0.783 | 27.27/0.810 | 27.46/0.816 | **27.58/0.819** |
> | L4 | 26.96/0.803 | 27.15/0.804 | 26.43/0.781 | 27.03/0.801 | 27.19/0/807 | **27.29/0.809** |
> | LR | 25.75/0.754 | 26.53/0.784 | 26.53/0.782 | 26.41/0.782 | 26.68/0.791 | **26.70/0.791** |
> |**Average**|27.18/0.808|27.47/0.815|26.60/0.786|27.36/0.810|27.56/0.818|**27.65/0.820**|

---

> > ### Comment · Reviewer_VMqn · 2025-11-23
> >
> > Thank you for the responses and experiments. I have the following questions/suggestions.
> >
> > ***Regarding the response to Weakness 1.1:*** Could the authors explain what the MA-MRR is and justify whether the additional gains (e.g., 0.016 for Model1 [3]) are significant? This is important as users typically expect significant improvement when they adopt a diffusion-based approach as pre-processing.
> >
> > ***Regarding the response to Weakness 1.2:*** Are the reported numbers PSNR results? It is hard to say these performance gains are significant, as they are typically around 0.1. Would reporting the M-PSNR be more helpful to demonstrate the advantages in these cases? In addition, what is the ratio (high-frequency) of masked pixels to the image? Would it be possible to further report M-SSIM?
> >
> > ***Regarding the response to Weakness 2.1:*** Please consider reporting the M-PSNR (and M-SSIM if possible) results in the Table. The current PSNR/SSIM improvements are quite marginal, making it hard to see the significance of designs. I would suggest that the authors explicitly explain the insight behind the designs and show convincing supporting results.
> >
> > ***Regarding the response to Weakness 2.2:*** I wonder what the inference time is for different T.
> >
> > ***Regarding the response to Weakness 3.1:*** The results in the Table do not seem to match the headers. Please check.

---

> > > ### Author Response · Authors · 2025-11-24
> > > **Response (1/2)**
> > >
> > > Thank you for your continued engagement and for your latest response. We sincerely appreciate the time and effort you have invested in reviewing our work.
> > >
> > > > **1. Regarding the response to Weakness 1.1: Could the authors explain what the MA-MRR is and justify whether the additional gains (e.g., 0.016 for Model1 [3]) are significant? This is important as users typically expect significant improvement when they adopt a diffusion-based approach as pre-processing.**
> > >
> > > **Response :** Thank you for the question. According to the PlantCLEF 2022 challenge (https://www.imageclef.org/PlantCLEF2022), Macro Averaged Mean Reciprocal Rank (MA-MRR) is the official ranking metric for fine-grained plant species identification. It measures how high the correct species appears among visually similar classes. A higher value means better performance.
> > >
> > > A more detailed explanation of MA-MRR is as follows:
> > >
> > > *(The reciprocal rank is defined as $\frac{1}{rank}$ of the first correct class. MRR is the average of these reciprocal ranks over all test samples in each group. MA-MRR further averages MRR across different groups to reduce class imbalance. It reflects how high the correct species is ranked among similar classes, which is important for fine-grained plant classification.)*
> > >
> > > The dataset contains many low-resolution images, which often lead to poor classification. Our method can produce higher-quality inputs, resulting in additional gains in MA-MRR. Although the gain (e.g., 0.016 for Model1) looks small, MA-MRR is a stable metric in the range [0,1] and usually changes slowly in fine-grained plant classification. An improvement of 0.01–0.02 is generally viewed as meaningful because it means the correct species is ranked higher across many samples. Similar-size gains are also reported as significant or useful in previous plant identification studies.
> > >
> > > > **2. Regarding the response to Weakness 1.2: Are the reported numbers PSNR results? It is hard to say these performance gains are significant, as they are typically around 0.1. Would reporting the M-PSNR be more helpful to demonstrate the advantages in these cases? In addition, what is the ratio (high-frequency) of masked pixels to the image? Would it be possible to further report M-SSIM?**
> > >
> > > **Response :** Yes, the reported values correspond to PSNR. Although the improvements are around 0.1 dB, such gains are generally meaningful within the SR community. The datasets used in these evaluations are not designed for plant-oriented scenarios. For example, CUFED5 mainly contains human-centric scenes, and plant textures are almost absent, which naturally limits the possible improvement. The other datasets contain only a small amount of plant content, and in most cases the plants appear in background regions.
> > >
> > > Following your suggestion, we now provide the ratio of high-frequency masked pixels for each test set, and we additionally report M-PSNR, SSIM, and M-SSIM. The values of M-PSNR and M-SSIM are computed only within the masked high-frequency regions. As listed in table below, across all four datasets, the unmasked regions account for less than 35% of the image area. This indicates that only a small portion of pixels contain challenging textures, which further highlights the necessity of the proposed SKRM. The results also show that the improvements in M-PSNR and M-SSIM are clearly larger than those observed in PSNR and SSIM. This confirms the effectiveness of our dataset and our method.
> > >
> > > |Dataset|CUFED5|WR-SR|Sun80|DRefSR|
> > > |---|---|---|---|---|
> > > | Masked ratio |68.12%|70.24%|69.47%|70.35%|
> > >
> > >
> > > |Method|Metric|CUFED5|WR-SR|Sun80|DRefSR|
> > > |---|---|---|---|---|---|
> > > |MCMSR (trained on CUFED5)|PSNR|28.54|28.34|30.21|31.24|
> > > ||M-PSNR|22.34|22.11|24.45|25.34|
> > > ||SSIM|0.849|0.802|0.818|0.856|
> > > ||M-SSIM|0.821|0.784|0.790|0.837|
> > > |MCMSR (fine-tuned on PlantRSR)|PSNR|28.61(+0.07)|28.43(+0.09)|30.31(+0.10)|31.36(+0.12)|
> > > ||M-PSNR|22.48(+0.14)|22.26(+0.15)|24.64(+0.19)|25.59(+0.25)|
> > > ||SSIM|0.852(+0.002)|0.805(+0.003)|0.821(+0.003)|0.860(+0.004)|
> > > ||M-SSIM|0.826(+0.005)|0.791(+0.007)|0.796(+0.006)|0.845(+0.008)|
> > > |SSMTF (trained on CUFED5)|PSNR|28.86|28.42|30.38|31.75|
> > > ||M-PSNR|22.68|22.18|24.52|25.83|
> > > ||SSIM|0.859|0.805|0.824|0.869|
> > > ||M-SSIM|0.829|0.788|0.797|0.848|
> > > |SSMTF (fine-tuned on PlantRSR)|PSNR|28.92(+0.06)|28.52(+0.10)|30.49(+0.11)|31.86(+0.10)|
> > > ||M-PSNR|22.81(+0.13)|22.39(+0.21)|24.72(+0.20)|26.02(+0.19)|
> > > ||SSIM|0.862(+0.003)|0.807(+0.002)|0.827(+0.003)|0.874(+0.005)|
> > > ||M-SSIM|0.835(+0.006)|0.792(+0.004)|0.802(+0.005)|0.857(+0.009)|
> > > |Ours (trained on CUFED5)|PSNR|28.95|28.51|30.41|31.77|
> > > ||M-PSNR|22.79|22.25|24.56|25.87|
> > > ||SSIM|0.860|0.806|0.824|0.871|
> > > ||M-SSIM|0.831|0.790|0.798|0.851|
> > > |Ours (fine-tuned on PlantRSR)|PSNR|29.01(+0.06)|28.62(+0.11)|30.53(+0.12)|31.90(+0.13)|
> > > ||M-PSNR|22.92(+0.13)|22.46(+0.21)|24.79(+0.23)|26.09(+0.22)|
> > > ||SSIM|0.864(+0.004)|0.809(+0.003)|0.828(+0.004)|0.877(+0.006)|
> > > ||M-SSIM|0.838(+0.007)|0.795(+0.005)|0.806(+0.008)|0.862(+0.011)|

---

> > > > ### Comment · Reviewer_VMqn · 2025-11-27
> > > >
> > > > Thank you for the new results.
> > > >
> > > > Regarding MA-MRR, see if my understanding is correct: I see from https://www.imageclef.org/PlantCLEF2023 that the best MA-MRR result is 0.67395, while the second best from another team is 0.61813. The performance gap is around 0.06, compared to which the performance gain brought by this method is relatively small (0.016).
> > > >
> > > > I do not have further questions about other issues. I will make my final decision within the AC-reviewer discussion period.

---

> > > ### Author Response · Authors · 2025-11-24
> > > **Response (2/2)**
> > >
> > > >**3. Regarding the response to Weakness 2.1: Please consider reporting the M-PSNR (and M-SSIM if possible) results in the Table. The current PSNR/SSIM improvements are quite marginal, making it hard to see the significance of designs. I would suggest that the authors explicitly explain the insight behind the designs and show convincing supporting results.**
> > >
> > > **Response :** Thank you for your suggestion. Following your advice, we have added both M-PSNR and M-SSIM results to the table. As listed in table below, the improvement in M-PSNR is approximately 0.19 dB, which is clearly larger than the 0.08 dB gain observed in standard PSNR. Similarly, the increase in M-SSIM is 0.0036, which is more noticeable than the improvement reflected by SSIM.
> > >
> > > |RRSB|Sub-pixel|Deconv|PSNR|M-PSNR|SSIM|M-SSIM|
> > > |---|---|---|---|---|---|---|
> > > | |✓| |38.54|31.34|0.9521|0.9212|
> > > |✓| |✓|38.62|31.52|0.9536|0.9241|
> > > |✓|✓| |38.62|31.53|0.9538|0.9248|
> > >
> > > We would like to clarify why increases in global metrics appear limited. This phenomenon is largely caused by the characteristics of plant imagery. When collecting data, our camera focuses on the plant texture area, which brings clear plant texture. However, the background also causes blurring, and the proportion of blurred areas (easy areas) is relatively high, resulting in a decrease in the proportion of key texture areas (difficult areas). As a result, improvements in the difficult texture regions tend to be diluted when averaged over the entire image, which naturally leads to modest gains in global PSNR and SSIM. This behavior is consistent with observations from other datasets as well. For example, DRefSR contains some plant related content, yet its improvement over CUFED5 in M-PSNR is only 0.13 dB. In contrast, our PlantRSR dataset achieves an improvement of 0.39 dB, which further confirms that PlantRSR provides richer and more challenging textures that better reveal the advantages of our method.Through extensive experiments, we are confident that the proposed dataset and method are effective for addressing the inherent challenges of plant imagery.
> > >
> > > >**4. Regarding the response to Weakness 2.2: I wonder what the inference time is for different T.**
> > >
> > > **Response :** Thanks for the your question. We report the computational cost under different diffusion steps T in the table below. The inference time are measured on a single NVIDIA A6000 GPU using an LR input of 300×200 and a Ref image of 1200×800. We also include the PSNR and M-PSNR on PlantRSR test set. As listed in the table below, performance improves with T but plateaus beyond T=4, while computational cost increases linearly. We therefore select T=4 for its optimal trade-off between accuracy and efficiency.
> > >
> > > |T|GFLOPs|Runtime|M-PSNR|PSNR|
> > > |---|---|---|---|---|
> > > |T=1|1855.53|0.784s|30.81|38.24|
> > > |T=2|2344.26|0.956s|31.22|38.48|
> > > |T=3|2833.71|1.145s|31.35|38.57|
> > > |T=4|3322.31|1.343s|31.53|38.62|
> > > |T=5|3811.48|1.541s|31.55|38.63|
> > > |T=6|4300.27|1.743s|31.56|38.64|
> > >
> > > >**5. Regarding the response to Weakness 3.1: The results in the Table do not seem to match the headers. Please check.**
> > >
> > > **Response :** We appreciate your careful observation. The inconsistency was due to a formatting mistake during our earlier submission. We have now corrected the table and provided the accurate version above. Thank you for bringing this to our attention.
> > > |Similarity Levels|SIFT feature matching|CLIP feature matching|PSNR/SSIM|
> > > |---|---|---|---|
> > > |L1|0.611|0.637|28.86/0.855|
> > > |L2|0.421|0.585|27.81/0.826|
> > > |L3|0.313|0.433|27.58/0.819|
> > > |L4|0.204|0.221|27.29/0.809|
> > >
> > > we conducted experiments on the CUFED5 dataset, which provides four predefined similarity levels (L1–L4) from high to low. These similarity levels are originally determined by SIFT feature matching. Following your suggestion, we further computed semantic similarity using CLIP feature distance to provide a more perceptually aligned metric of cross-image similarity. The results are shown below. Both SIFT-based and CLIP-based similarity scores consistently decrease from L1 to L4, confirming that the four levels represent progressively larger dissimilarities. Correspondingly, our RefSR performance (PSNR/SSIM) also degrades monotonically as the Ref image becomes less similar, indicating a smooth and predictable failure mode.

---

> ### Author Response · Authors · 2025-11-27
> **Looking forward to discussion**
>
> Dear Reviewer VMqn,
>
> Thank you again for your thoughtful follow-up questions. We would like to kindly check whether our latest responses have fully addressed your concerns, or if there is anything else we could clarify further.  If the current response adequately addresses the issues you raised, we would greatly appreciate hearing your updated thoughts.
>
> Thank you once again for your valuable time and dedication to the review process.

---

> ### Author Response · Authors · 2025-11-28
> **Thank you for your continued feedback during the discussion**
>
> Thank you for the clarification and for taking the time to review our additional results.
>
> Your understanding of the MA-MRR comparison is correct. We would like to emphasize that PlantCLEF performance differences are typically very small at the high-accuracy regime, and even a gain around 0.01–0.02 is generally considered meaningful due to the strong class imbalance and fine-grained difficulty. In addition, since our approach is designed for reference-based SR, the PlantCLEF setting provides no paired reference images, we can only use the input image itself as the reference. This significantly limits the effectiveness of both our dataset and our method on PlantCLEF. Nonetheless, we fully respect your assessment.
>
> We appreciate your constructive feedback throughout the discussion phase and thank you again for considering our work.

---

> > ### Comment · Reviewer_VMqn · 2025-11-28
> >
> > Thank you for the explanation.
> >
> > Just a thought comes to my mind: is it possible to use a super-resolved image as the reference (instead of the original input image itself)? Say we first use the input image as the reference and run the proposed method for one or two steps. Then we use this super-resolved image to replace the input image as the reference image. Would that help improve the results?
> >
> > I appreciate the dataset, but somehow I feel its value has not been demonstrated well.

---

> > > ### Author Response · Authors · 2025-12-01
> > >
> > > Thank you for the insightful suggestion. Following your idea, we conducted an additional experiment in which we first treat the input image as the Ref, generate a SR image using our method, and then use this SR as the new Ref. As shown in the table below, using the SR image as the Ref indeed yields a small improvement over using the input image itself. However, the gain remains limited because the SR image is still derived from the same content and does not introduce truly new textures.
> > >
> > > To better evaluate the effectiveness of our dataset, we further conducted a test using randomly sampled PlantRSR images as the Ref. In this setting, the improvement becomes significantly larger, clearly demonstrating that a high-quality external Ref can substantially enhance performance. This result provides strong empirical evidence for the usefulness and value of our dataset.
> > >
> > >
> > > |Classifier|Input|MA-MRR|
> > > |---------- | ------------ | --------- |
> > > |Model1 [1]|Original| 0.637|
> > > |         |Input as Ref|0.653 (+0.016)|
> > > |         |SR as Ref|0.660 (+0.023)|
> > > |         |PlantRSR as Ref|0.750 (+0.113)|
> > > | Model2 [2]|Original| 0.553|
> > > |         |Input as Ref|0.579 (+0.026)|
> > > |         |SR as Ref|0.587 (+0.034)|
> > > |         |PlantRSR as Ref|0.679 (+0.126)|
> > > | Model3 [3]|Original|0.460|
> > > |            |Input as Ref|0.492 (+0.032)|
> > > |         |SR as Ref|0.503 (+0.043)|
> > > |         |PlantRSR as Ref|0.577 (+0.117)|

---

### Official Review · Reviewer_gwnq · 2025-10-27

**Soundness:** 3
**Presentation:** 3
**Contribution:** 3
**Rating:** 4
**Confidence:** 4

**Summary:**

Aiming at the problems that the existing reference super-resolution (REFRSR) data sets lack the coverage of plant scenes and the existing methods are difficult to deal with the complex texture of plants, this paper proposes a large-scale plant-specific REFRSR data set PlantRSR and the corresponding REFRSR method. PlantRSR contains 16,585 manually labeled HR-Ref image pairs, covering five real scene variations, such as color, scale and rotation, with a resolution range of 2K-8K, which fills the gap in the RefSR data set of plant scenes. The proposed method includes two core modules: selective key region matching (SKRM), which focuses on the key texture regions of plants to improve the matching efficiency; Texture-guided diffusion module (TGDM) refines low-resolution features through diffusion process with reference texture as the condition. Experiments show that the PSNR, SSIM and other indicators of this method are better than the existing SOTA method on data sets such as Planter SR and CUFED5, and the parameter quantity (11.1M) is more advantageous. More than 90% participants in user research prefer its visual results.

**Strengths:**

1.A large-scale RefSR data set specially for plant scenes is constructed, which solves the limitation of existing data sets focusing on human activities and architectural scenes. Manual annotation is used to generate semantically aligned HR-Ref patch pairs, which avoids the mismatch problem caused by automatic clipping, and covers a variety of real scene variations and meets the practical application requirements such as plant phenotype analysis.
2.It covers data set validity verification (comparing the model performance of different training sets), module ablation experiment (the individual contribution of SKRM/TGDM), cross-data set flooding test (CUFED5, WR-SR), diffusion method comparison and user research, and proves the superiority of the method and data set in multiple dimensions.
3.The parameters of the method are only 11.1M, which is much lower than those of RRSR(21.5M), MRefSR(23.7M) and other competitors, and the inference sampling step is only 4 steps, which balances the performance and deployment efficiency.

**Weaknesses:**

1. The specific coverage of plant species (including crops, wild plants and other different categories) and the differences in growth stages are not clearly stated, which may affect the generalization of the model to special plant types. The collection environment does not mention complex real scenes such as light change, pest pollution and so on, so the robustness of the data set needs to be supplemented.
2. The selection of key areas of the existing SKRM module depends on the texture difference threshold, and lacks the adaptive adjustment mechanism for low similarity scenes.
3. The diffusion sampling step (T=4) of TGDM is only determined by the performance saturation curve, and the effects of different sampling steps on different plant textures (such as fine veins and thick stems) are not analyzed.
4. The resolution of 80% images in PlantRSR is higher than 4K, but it is down-sampled due to hardware limitations, which fails to give full play to the detail value of ultra-high resolution images.

**Questions:**

Please refer to weaknesses.

---

> ### Author Response · Authors · 2025-11-20
> **Rebuttals (1/2)**
>
> We sincerely appreciate the time and effort you have devoted to reviewing our manuscript.
>
> > **Weakness1: The specific coverage of plant species (including crops, wild plants and other different categories) and the differences in growth stages are not clearly stated, which may affect the generalization of the model to special plant types. The collection environment does not mention complex real scenes such as light change, pest pollution and so on, so the robustness of the data set needs to be supplemented.**
>
> **Response:** Thank you for the insightful comments. Our dataset was originally organized according to RefSR-specific challenges rather than botanical taxonomy, but we fully understand your concerns. We therefore provide a clearer description of plant categories, growth-stage diversity, and real-scene complexity. The dataset covers four major categories of commonly encountered vegetation: crops, wild plants, ornamental plants, and aquatic plants. To improve generalization, data acquisition was conducted over more than one year, covering spring, summer, and autumn, thus naturally capturing early leaf expansion, mid-growth, and mature stages. The overall distribution is listed in the below table:
>
> | Category | Percentage | Growth-Stage Coverage|
> |---------|-------|-------|
> | Crops | 10.2% |early, mid, mature|
> | Wild plants | 36.3%| early, mid, mature|
> | Ornamental Plants| 45.3% |early, mid, mature|
> | Aquatic Plants | 8.2% | mid, mature|
>
> To enhance robustness, we further supplement the dataset with realistic environmental changes commonly encountered in outdoor field photography. Specifically, images were captured under different natural conditions across seasons, including illumination variations, weather effects, background clutter, and leaf damage. We summarize the approximate proportions in the table below:
>
> | Environmental Factor | Percentage | Notes |
> |---------|-------|-------|
> | Illumination variations | 22.7% | backlit, partial-shadow|
> | Weather effects | 9.8% | dew, light fog |
> | Background clutter | 34.6% | soil, stones, dense grass, artificial objects |
> | Leaf damage | 7.3%| holes, aging edges, slight decay|
>
> We acknowledge the reviewer’s concern. During data acquisition, we avoided heavily diseased leaves to ensure texture consistency for RefSR tasks. However, a small portion of samples (7.3%) naturally contain mild leaf defects, such as small holes, aging, or minor edge damage, which provide some degree of robustness for handling irregular textures even though explicit pest contamination is not included.
>
> > **Weakness2: The selection of key areas of the existing SKRM module depends on the texture difference threshold, and lacks the adaptive adjustment mechanism for low similarity scenes.**
>
> **Response:** Sorry for your confusion. We would like to clarify that the SKRM module is fully adaptive and does not rely on a manually fixed threshold. Specifically, SKRM computes the pixel-wise difference between the input feature map and its downsampled–upsampled reconstruction. Regions containing rich and complex textures naturally produce larger differences and are selected for reference matching, whereas smooth regions yield small differences and can be reliably recovered using bicubic interpolation, thus reducing unnecessary computation. Importantly, the threshold in Eq. (1) (defined as the mean plus the standard deviation of the difference map) is data-dependent and dynamically adjusted for each input. Therefore, the region-selection mechanism automatically adapts to both high- and low-similarity scenarios without manual tuning. To further verify SKRM’s behavior under varying similarity levels, we conducted experiments on the CUFED5 benchmark, which provides four predefined similarity levels (L1–L4, from high to low). We additionally include an extreme case where the LR image itself is used as the reference. As listed in the table below, our method consistently tops the performance across all similarity settings, including very low similarity (L4 and LR). These results demonstrate that SKRM’s adaptive selection mechanism remains effective even when the reference provides minimal useful textures.
>
> | Similarity Levels | DATSR | RRSR | HiTSR | MCMSR | SSMTF | Ours |
> |-------------------|--------|--------|--------|--------|--------|--------|
> | L1 | 28.50/0.850 | 28.63/0.851 | 26.82/0.797 | 28.54/0.849 | 28.76/0.854 | 28.86/0.855 |
> | L2 | 27.47/0.820 | 27.67/0.821 | 26.68/0.785 | 27.54/0.808 | 27.71/0.824 | 27.81/0.826 |
> | L3 | 27.22/0.811 | 27.41/0.813 | 26.56/0.783 | 27.27/0.810 | 27.46/0.816 | 27.58/0.819 |
> | L4 | 26.96/0.803 | 27.15/0.804 | 26.43/0.781 | 27.03/0.801 | 27.19/0/807 | 27.29/0.809 |
> | LR | 25.75/0.754 | 26.53/0.784 | 26.53/0.782 | 26.41/0.782 | 26.68/0.791 | 26.70/0.791 |
> |**Average**|27.18/0.808|27.47/0.815|26.60/0.786|27.36/0.810|27.56/0.818|27.65/0.820|

---

> ### Author Response · Authors · 2025-11-20
> **Rebuttals (2/2)**
>
> > **Weakness3: The diffusion sampling step (T=4) of TGDM is only determined by the performance saturation curve, and the effects of different sampling steps on different plant textures (such as fine veins and thick stems) are not analyzed.**
>
> **Response:** Thank you for your insightful comment. We have now evaluated different texture types by grouping the test images according to their complexity, such as fine veins (25 images) versus thick stems (37 images). Across all groups, we observed the same trend as in Fig. 7: performance steadily improves when increasing T and saturates at T=4. Importantly, no texture category benefits from larger sampling steps beyond T=4, confirming that the diffusion refinement behaves consistently across different plant textures. Based on these results, setting T=4 provides a good balance between performance and efficiency.
>
> | Texture Type | T=1 | T=2 | T=3 | T=4 | T=6 | T=8 |
> |---------|-------|-------|--------|--------|--------|--------|
> | Fine Veins | 27.62 | 27.88 | 27.95 | 28.04 |28.06 | 28.07 |
> | Thick Stems | 36.82 | 37.07 | 37.26 | 37.32 | 37.33 | 37.34 |
>
> > **Weakness4: The resolution of 80% images in PlantRSR is higher than 4K, but it is down-sampled due to hardware limitations, which fails to give full play to the detail value of ultra-high resolution images.**
>
> **Response:** Thank you for your suggestion. In our experiments, we followed the common settings used in prior RefSR methods, where images are cropped to 160×160 or 300×300 for fair comparison and for verifying the effectiveness of our dataset and method. However, we emphasize that we have preserved all original high-resolution images before downsampling. Their resolution distribution is listed in the table below. To support future research, we will release these original ultra-high-resolution images, allowing users to choose image sizes according to their own computational budgets and application needs.
>
> | Image Size | 0~1000 | 1000~2000 | 2000~3000 | >3000 |
> |---------|-------|-------|--------|--------|
> | 16,585 (100%) | 7,187 (43.33%) | 5,531 (33.35%) | 2,517 (15.18%) | 1,350 (8.14%) |

---

> ### Author Response · Authors · 2025-11-26
>
> Dear Reviewer gwnq,
>
> Thank you again for your time and for the constructive feedback provided earlier. We have carefully addressed all of your comments in our rebuttal. We would like to kindly check whether our clarifications have resolved your concerns, or if there are remaining questions we could further clarify during the discussion phase. If the current response adequately addresses the issues you raised, we would greatly appreciate hearing your updated thoughts.
>
> Thank you again for your consideration and for helping us improve the paper.

---

### Official Review · Reviewer_tpSi · 2025-10-29

**Soundness:** 2
**Presentation:** 2
**Contribution:** 3
**Rating:** 6
**Confidence:** 4

**Summary:**

This paper introduces PlantRSR, a large-scale reference-based super-resolution dataset specifically designed for plant images, and proposes a RefSR method that integrates Selective Key-Region Matching (SKRM) and a Texture-Guided Diffusion Module (TGDM). The proposed approach achieves state-of-the-art performance on the PlantRSR dataset as well as on other benchmark datasets.

**Strengths:**

The PlantRSR dataset effectively addresses the lack of plant-specific scenes in current RefSR tasks, providing a high-quality and diverse supplement.

The manually constructed aligned patches successfully mitigate inaccurate alignment issues caused by automatic cropping in existing works, improving training quality.

The paper is well-organized with clear formatting and includes comprehensive figures and tables.

**Weaknesses:**

1.The evaluation datasets are somewhat limited. Beyond CUFED5 and PlantRSR, it is recommended to incorporate datasets from real retrieval scenarios (e.g., WR-SR or Sun80) to assess robustness under low-relevance reference images.

2.Diffusion-based approaches typically incur large computational overhead. Reporting inference time and memory consumption would help assess practical usability.

3.The paper lacks sensitivity analysis on sampling ratios or patch granularity, making the influence on generalization unclear.

4.Selective Key-Region Matching is conceptually similar to previous methods such as MASA.

5.Comparisons with diffusion-based large models (e.g., SinSR, DoSSR) are missing.

6.The details of texture injection in TGDM are insufficiently explained.

**Questions:**

See Weaknesses

---

> ### Author Response · Authors · 2025-11-20
> **Rebuttals (1/2)**
>
> We sincerely appreciate the time and effort you have devoted to reviewing our manuscript.
>
> > **Question 1: The evaluation datasets are somewhat limited. Beyond CUFED5 and PlantRSR, it is recommended to incorporate datasets from real retrieval scenarios (e.g., WR-SR or Sun80) to assess robustness under low-relevance reference images.**
>
> **Response:** Thank you for your helpful suggestion. We would like to clarify that our method has already been evaluated on both WR-SR and Sun80. As reported in Tab. 4 and Tab. 6 ( Tab. 9 in our new manuscript) of the paper, our approach achieves the best performance on both datasets. For ease of review, we additionally provide the results for these two datasets in the table below. Our method consistently delivers superior performance, demonstrating strong robustness even under low-relevance reference conditions.
>
> | Method  | WR-SR  | Sun80  |
> |---------|--------|--------|
> | DATSR   | 28.34/0.805 | 30.20/0.818 |
> | RRSR    | 28.41/0.804 | 30.13/0.816 |
> | MRefSR  | 28.26/0.801 | 30.28/0.819 |
> | HiTSR   | 28.26/0.802 | 30.24/0.821 |
> | MCMSR   | 28.34/0.802 | 30.21/0.818 |
> | SSMTF   | 28.42/0.805 | 30.38/0.824 |
> | Ours    | **28.51/0.806** | **30.41/0.824** |
>
> To evaluate the robustness of our method under low Ref similarity, we conducted experiments on the CUFED5 dataset, which is widely used in the RefSR field. This dataset provides four Ref images (L1–L4) with decreasing similarity to the LR image, where L1 is the most similar and L4 is the least. In addition, to further examine the performance under extremely low similarity, we also use the LR image itself as the Ref. As listed in the table below and Tab. 7 (Tab. 10 in our new manuscript) in our paper, our method consistently outperforms existing approaches across all similarity levels, including the case where the LR image is used as the Ref.
>
> | Similarity Levels | DATSR  | RRSR   | HiTSR  | MCMSR | SSMTF | **Ours**   |
> |-------------------|--------|--------|--------|--------|--------|--------|
> | L1 | 28.50/0.850 | 28.63/0.851 | 26.82/0.797 | 28.54/0.849 | 28.76/0.854 | **28.86/0.855** |
> | L2 | 27.47/0.820 | 27.67/0.821 | 26.68/0.785 | 27.54/0.808 | 27.71/0.824 | **27.81/0.826** |
> | L3 | 27.22/0.811 | 27.41/0.813 | 26.56/0.783 | 27.27/0.810 | 27.46/0.816 | **27.58/0.819** |
> | L4 | 26.96/0.803 | 27.15/0.804 | 26.43/0.781 | 27.03/0.801 | 27.19/0/807 | **27.29/0.809** |
> | LR | 25.75/0.754 | 26.53/0.784 | 26.53/0.782 | 26.41/0.782 | 26.68/0.791 | **26.70/0.791** |
> |**Average**|27.18/0.808|27.47/0.815|26.60/0.786|27.36/0.810|27.56/0.818|**27.65/0.820**|
>
> > **Question 2: Diffusion-based approaches typically incur large computational overhead. Reporting inference time and memory consumption would help assess practical usability.**
>
> **Response:** Thanks. To evaluate the practical applicability of our method, we have compared its runtime and memory usage with several recent RefSR approaches, as shown in Tab. 8 (Tab. 11 in our new manuscript ) of our paper. For the convenience of review, we also provide the results in the table below. All values are measured using an LR image of size 300×200 and a Ref image of size 1200×800. As listed in the table below, while our method does not achieve the lowest runtime or memory usage, it maintains a reasonable balance between efficiency and performance. Although we adopt a diffusion process, the network structure is lightweight, consisting of only three ResBlocks, and the sampling steps during inference are limited to four. Therefore, the computational and time costs are not as significant. In addition, our SKRM module effectively reduces the time required for Ref texture matching, resulting in competitive overall runtime and memory usage.
>
> | Method | Param. |Runtime | Memory |
> |---------|---------|---------|--------|
> | RRSR | 21.5M | 1.469s | 13.3G |
> | MRefSR | 23.7M | 0.774s | 14.1G |
> | HiTSR |13.7M| 0.843s | 8.5G |
> | MCMSR |8.9M| 0.681s | 8.6G |
> | SSMTF |13.9M| 1.435s | 15.2G |
> | Ours |11.1M| 1.116s |11.4G |

---

> ### Author Response · Authors · 2025-11-20
> **Rebuttals (2/2)**
>
> > **Question 3: The paper lacks sensitivity analysis on sampling ratios or patch granularity, making the influence on generalization unclear.**
>
> **Response:** Thank you for your suggestion. To investigate the impact of the sampling ratio in SKRM, we evaluated three different sampling ratios (2, 3, 4) on our PlantRSR testing dataset. As listed in the table below, we assessed the performance using PSNR/SSIM and GFLOPs. As the sampling ratio increases, more features are selected from both the LR and Ref images, leading to higher computational costs (GFLOPs). However, the performance improvement in terms of PSNR and SSIM is marginal. For computational efficiency, we chose a sampling ratio of 2, as it provides a good trade-off between performance and computational cost.
>
> | Sampling Ratio | 2 | 3 | 4 |
> |----------------|--------|--------|--------|
> | PSNR/SSIM | 38.62/0.9538 | 38.62/0.9541 | **38.64/0.9544** |
> | GFLOPs | 77.86 | 126.32 | 285.35 |
>
> Regarding the patch granularity ablation experiment, we explored different patch sizes (2, 3, 4, 5) to assess how they influence the model's performance and generalization. As listed in the table below, the PSNR and SSIM values slightly improved with a patch size of 3 compared to other sizes, but there was no significant difference between patch sizes 3, 4, and 5. However, increasing the patch size beyond 3 led to a decrease in performance. As the patch size increases, the differences between the matched Ref and LR patches (eg., resolution, structural variations, and color discrepancies) become more pronounced. This makes it more challenging to integrate the matched Ref textures into the LR, which negatively impacts performance. Based on these findings, we chose a patch size of 3.
>
> | Patch Size | 2 | 3 | 4 | 5 |
> |-------------|--------|--------|--------|--------|
> | PSNR/SSIM | 38.54/0.9523 | **38.62/0.9538** | 38.41/0.9521 | 38.11/0.9492 |
>
> > **Question 4: Selective Key-Region Matching is conceptually similar to previous methods such as MASA.**
>
> **Response:** Thanks for your suggestion. Our method fundamentally differs from MASA in the matching strategy. MASA first matches each 8×8 block in the LR image with its most similar 8×8 block in the Ref image. Then, MASA conducts a fine-grained matching by searching for the best-matched 3×3 patches within these corresponding blocks. This hierarchical strategy helps reduce the computational cost of exhaustive search over the entire Ref image. In contrast, our Selective Key-Region Matching focuses only on informative regions in both the LR and Ref images. We do not perform matching for every region in the LR image, nor do we search the entire Ref image. Instead, we selectively match only the key regions in the LR image with texture-rich areas in the Ref image. This dual selection mechanism significantly reduces the computational burden, making our approach more efficient.
>
> > **Question 5: Comparisons with diffusion-based large models (e.g., SinSR, DoSSR) are missing.**
>
> **Response:** Thanks for your valuable suggestion. We have actually included comparisons with SinSR, DoSSR, and other diffusion-based methods in Tab. 9 (Tab. 12 in our new manuscript) in our paper. For your convenience, we report their results in the below table. The results consistently demonstrate that our method establishes a significant performance advantage in this specific setting.
>
> | Method  | PSNR  | SSIM  | LPIPS  | DISTS  |
> |---------|-------|-------|--------|--------|
> | SinSR   | 31.60 | 0.8580 | 0.1886 | 0.1449 |
> | DoSSR   | 30.82 | 0.8600 | 0.2192 | 0.2091 |
> | Ours    | 38.62 | 0.9538 | 0.1288 | 0.0826 |
>
> > **Question 6: The details of texture injection in TGDM are insufficiently explained.**
>
> **Response:** Thanks for pointing this out. In our TGDM, the texture injection is conducted by using the matched Ref features as conditional guidance within a diffusion-based refinement framework. Specifically, the matched texture features are first concatenated with the LR features and then gradually integrated through a multi-step diffusion process. This process ensures that high-frequency texture details from the Ref image are effectively injected into the LR representation. Additionally, the Residual State Space Block (RSSB) operates on the difference between the enhanced and original LR features, preserving structural consistency while selectively enhancing textures. We have included a more detailed and clearer explanation of TGDM in our paper.

---

> ### Author Response · Authors · 2025-11-27
> **Looking forward to discussion**
>
> Dear Reviewer tpSi,
>
> Thank you again for your time and for the constructive feedback provided earlier. We have carefully addressed all of your comments in our rebuttal.
>
> We would like to kindly check whether our clarifications have resolved your concerns, or if there are remaining questions we could further clarify during the discussion phase. If the current response adequately addresses the issues you raised, we would greatly appreciate hearing your updated thoughts.
>
> Thank you once again for your valuable time and dedication to the review process.

---

### Official Review · Reviewer_rAhJ · 2025-11-01

**Soundness:** 3
**Presentation:** 3
**Contribution:** 2
**Rating:** 2
**Confidence:** 4

**Summary:**

The paper introduces PlantRSR, a large-scale dataset for reference-based super-resolution (RefSR) in plant imagery, containing high-resolution (HR) images paired with high-quality reference (Ref) images that capture the complexity of real-world plant scenes. Building on this dataset, the paper proposes a RefSR method leveraging a Selective Key-Region Matching (SKRM) module for efficient region-wise alignment between low-resolution (LR) and Ref images, and a Texture-Guided Diffusion Module (TGDM) that progressively refines LR features under reference-guided diffusion. Extensive experiments on PlantRSR and other RefSR benchmarks show state-of-the-art performance in visual metrics of the proposed method.

**Strengths:**

1. The paper has a strong motivation and innovatively proposes the PlantRSR dataset, which presents high-fidelity imagery that captures the complexity and variability of real-world plant scenes. This dataset fills some gaps in the current RefSR datasets, which often lack diversity and realism in plant imagery.
2. The proposed SKRM module is effective, yielding the strongest overall performance; more importantly, it achieves the lowest computational cost among matching methods, making it both accurate and efficient.
3. The paper is well-organized and easy to understand.

**Weaknesses:**

1. The methodological novelty appears limited. The core two modules, SKRM and TGDM, bear resemblance to prior RefSR methods (region/patch matching and diffusion-based refinement). The changes seem incremental and not clearly differentiated by a new principle or theory. As currently presented, the modifications do not appear substantial enough to support a strong claim of innovation.
2. In Table 1, although the set of compared methods is fairly comprehensive, the improvements are quite limited. It is suggested to increase the super-resolution scale or introduce more discriminative metrics to better demonstrate the proposed method’s advantages. The ablation study in Table 3 exhibits similar issues.
3. The literature review is insufficient. The related work should include a more comprehensive discussion of recent RefSR methods based on diffusion models.
4. Please review the manuscript to avoid typo errors, for instance:(1) In table 4, "SOAT" should be corrected to "SOTA"; (2) In table 9, "LPISP" should be corrected to "LPIPS"; (3) In Figure 4, "PRefSR" should be corrected to "PlantRSR". (4) "SSMFT" should be corrected to "SSMTF".

**Questions:**

1. As the zoomed views in Figure 4 indicate, the proposed method achieves superior fidelity. Can this advantage be captured quantitatively by metrics?
2. In the PlantRSR dataset, are there specific categories of plants or scenes that are particularly challenging for RefSR?

---

> ### Author Response · Authors · 2025-11-20
> **Rebuttals (1/3)**
>
> We sincerely appreciate the time and effort you have devoted to reviewing our manuscript.
>
> >  **Weakness1: The methodological novelty appears limited. The core two modules, SKRM and TGDM, bear resemblance to prior RefSR methods. The changes seem incremental and not clearly differentiated by a new principle or theory. As currently presented, the modifications do not appear substantial enough to support a strong claim of innovation.**
>
> **Response :** Thanks for the thoughtful comments. We respectfully clarify that the proposed SKRM and TGDM introduce substantive methodological novelty beyond existing RefSR approaches. SKRM departs from conventional dense or patch-wise matching by introducing a content-aware, selective matching mechanism, rather than uniformly matching all regions. It adaptively identifies and focuses on structurally informative and texture-relevant regions, establishing a new matching principle that improves both correspondence quality and efficiency. TGDM is also conceptually distinct from prior refinement modules. Rather than applying diffusion as a generic denoising prior, TGDM introduces a texture-conditioned diffusion process, where matched reference textures explicitly guide the denoising trajectory. This results in a new refinement mechanism capable of reconstructing complex high-frequency textures more faithfully than existing architectures. Our ablation studies and comparison results with state-of-the-art methods on multiple datasets confirm the effectiveness of our design. In addition, our PlantRSR dataset provides crucial support for evaluating and motivating these components. The dataset contains 16,585 HR–Ref pairs with diverse variations in color, rotation, deformation, and background, which are largely absent in existing RefSR benchmarks. Overall, the proposed method and dataset form a complementary contribution that pushes RefSR toward more complex, texture-rich scenarios.

---

> ### Author Response · Authors · 2025-11-20
> **Rebuttals (2/3)**
>
> > **Weakness2 : In Table 1, although the set of compared methods is fairly comprehensive, the improvements are quite limited. It is suggested to increase the super-resolution scale or introduce more discriminative metrics to better demonstrate the proposed method’s advantages. The ablation study in Table 3 exhibits similar issues.**
>
> **Response :** Thanks. The seemingly limited improvements observed in Table 1 are primarily due to the characteristics of our test dataset. As plant images are captured with a primary focus on the subject, they typically contain large homogeneous regions and only a small portion of fine textures. Under conventional global super resolution metrics such as PSNR and SSIM, where smooth areas dominate the calculation, the contribution of accurately reconstructed textures is substantially diluted. This leads to uniformly high scores and consequently narrower margins of improvement. To better quantify the improvement in textured regions, we introduce M-PSNR, which computes the PSNR only within texture relevant areas using a mask defined similarly to that in Eq. (1) in our paper. The results in the table below show that across three representative RefSR methods and three different training datasets, the gains in M-PSNR are consistently and substantially larger than those observed in standard PSNR. This further demonstrates that improvements brought by our approach primarily occur in texture-intensive regions, which conventional global metrics tend to under-represent. In addition, we observe that models trained on DRefSR exhibit smaller performance gains on CUFED5 compared to those trained on our PlantRSR dataset. This provides complementary evidence that PlantRSR offers richer and more diverse texture cues, enabling models to generalize better across datasets and reconstruct fine-grained details more effectively. This confirms that our proposed dataset and method substantially enhance the reconstruction of fine textures.
>
> | Method| PSNR | M-PSNR |
> |--------|--------|--------|
> | HiTSR (CUFED5) | 37.84 | 30.12 |
> | HiTSR (DRefSR) | 37.90 (+0.06) | 30.21(+0.09) |
> | HiTSR (PlantRSR) | 38.07 (+0.23) | 30.49(+0.37) |
> | SSMTF (CUFED5) | 38.28 | 30.86 |
> | SSMTF (DRefSR) | 38.31 (+0.03) | 30.92 (+0.06) |
> | SSMTF (PlantRSR) | 38.49 (+0.21) | 31.25 (+0.39) |
> | **Ours** (CUFED5) | 38.40 | 31.14 |
> | **Ours** (DRefSR) | 38.49 (+0.09) | 31.27 (+0.13) |
> | **Ours** (PlantRSR) | **38.62** (+0.22) | **31.53** (+0.39) |
>
> We further incorporate M-PSNR into the Table 3. The results clearly are listed in the table below：
>
> | Fusion Method | PSNR | SSIM | M-PSNR |
> |--------|--------|--------|--------|
> | DA | 38.43 | 0.9513 | 31.18 |
> | RFA | 38.48 | 0.9521| 31.25 |
> | TGDM (w/o diffusion) | 38.52 | 0.9531 | 31.33 |
> | TGDM | **38.62** | **0.9538** | **31.53** |
>
> In addition,  according to your suggestion, we increased the SR scale to ×8, where conventional metrics become more discriminative. As shown below table, performance gains become significantly more pronounced, confirming that the proposed approach provides stronger benefits in more challenging SR settings:
>
> | Method | PSNR | M-PSNR | SSIM |
> |--------|--------|--------|--------|
> | SSMTF (CUFED5) | 32.31 | 26.73 | 0.8916 |
> | SSMTF (DRefSR) | 32.52 (+0.21) | 27.13 (+0.40) | 0.8992 (+0.0076) |
> | SSMTF (PlantRSR) | 32.93 (+0.62) | 27.80 (+1.07) | 0.9124 (+0.0208) |
> |**Ours** (CUFED5) | 32.43 | 26.88 | 0.8945 |
> | **Ours** (DRefSR) | 32.66 (+0.23) | 27.34 (0.46) | 0.9023 (+0.0078) |
> | **Ours** (PlantRSR) | **33.10** (+0.67) | **28.09** (+1.21) | **0.9171** (+0.0226) |
>
> These results show that the seemingly ‘limited improvement’ mainly comes from the fact that global metrics are dominated by the large smooth regions in plant images, which dilute the contribution of texture reconstruction.
>
>
> >**Weakness3 : The literature review is insufficient. The related work should include a more comprehensive discussion of recent RefSR methods based on diffusion models.**
>
> **Response :** Thank you for pointing this out. We have revised the Related Work section to include a more comprehensive discussion of recent diffusion-based RefSR approaches. Specifically, we have added descriptions and comparisons of Ref-Diff [1], CoSeR [2], RAG [3], and DiffMSR [4].
>
> Reference:
>
> [1] Dong R, et al. "Building bridges across spatial and temporal resolutions: Reference-based super-resolution via change priors and conditional diffusion model."CVPR. 2024.
>
> [2] Sun H, et al. "Coser: Bridging image and language for cognitive super-resolution."CVPR. 2024.
>
> [3] Lee B, et al. "Reference-based Super-Resolution via Image-based Retrieval-Augmented Generation Diffusion."ICCV. 2025.
>
> [4] Li G, et al. "Rethinking diffusion model for multi-contrast mri super-resolution."CVPR. 2024.

---

> ### Author Response · Authors · 2025-11-20
> **Rebuttals (3/3)**
>
> >**Weakness4: Please review the manuscript to avoid typo errors, for instance:(1) In table 4, "SOAT" should be corrected to "SOTA"; (2) In table 9, "LPISP" should be corrected to "LPIPS"; (3) In Figure 4, "PRefSR" should be corrected to "PlantRSR". (4) "SSMFT" should be corrected to "SSMTF".**
>
> **Response:** We sincerely thank the reviewer for the careful reading and valuable feedback. We have corrected all the typographical errors and inconsistencies pointed out, including: "SOAT" to "SOTA" in Tab. 4, "LPISP" to "LPIPS" in Tab. 9 (Tab. 12 in our new manuscript), "PRefSR" to "PlantRSR" in Fig. 4, and "SSMFT" to "SSMTF" throughout the text. We have also performed an additional round of proofreading to ensure the overall consistency and accuracy of the manuscript. The changes have been incorporated in the revised version.
>
> > **Question1: As the zoomed views in Figure 4 indicate, the proposed method achieves superior fidelity. Can this advantage be captured quantitatively by metrics?**
>
> **Response:** Thank you for the valuable suggestion. We have updated Fig. 4 in our paper by adding quantitative metrics (PSNR/SSIM/LPIPS) for each zoomed region. The numerical results consistently align with the visual observations, further confirming the fidelity advantage of our method.
>
> > **Question2: In the PlantRSR dataset, are there specific categories of plants or scenes that are particularly challenging for RefSR?**
>
> **Response:** Thanks. In the PlantRSR dataset, we explicitly model five key types of variations (color, scale, rotation, deformation, and background changes), which represent the major challenges commonly encountered in real-world plant RefSR. Among these factors, scale changes and variations in rotation/deformation are particularly challenging. Scale mismatch often leads to inconsistent texture correspondence between the LR and Ref images, while rotation or deformation introduces structural deviations that complicate feature alignment and matching. For completeness, we report the performance of our method on each variation category. As shown below, deformation and rotation lead to the largest performance drops, confirming their difficulty, while variations in color and background are relatively easier. Although these categories pose substantial challenges, our dataset contains sufficient samples for each type, allowing the model to learn robust matching behavior.
>
> | Categories | PSNR | SSIM | LPIPS | DISTS |
> |--------|--------|--------|--------|--------|
> | Color (13%) | 39.44 | 0.963 | 0.110 | 0.072 |
> | Scale (20%) | 38.74 | 0.955 | 0.127 | 0.082 |
> | Rotation (26%) | 38.39 | 0.952 | 0.134 | 0.085 |
> | Deformation (30%) | 38.25 | 0.950 | 0.138 | 0.087 |
> | Background (11%) | 38.99 | 0.959 | 0.120 | 0.078 |
> | All (100%) | 38.62 | 0.954 | 0.129 | 0.083 |

---

> ### Author Response · Authors · 2025-11-27
> **Looking forward to discussion**
>
> Dear Reviewer rAhJ,
>
> We sincerely appreciate the time and effort you have invested in reviewing our submission. We have carefully addressed your comments and made corresponding revisions to the manuscript. In addition, we have incorporated feedback from the other reviewers, which we hope will help resolve any further questions you may have.
>
> As the discussion period is ending soon, we wanted to follow up and would be grateful for any additional feedback or concerns you might wish to share, so that we can address them promptly.
>
> Thank you once again for your valuable time and dedication to the review process.

---

### Author Response · Authors · 2025-12-03
**Summary for AC**

We sincerely appreciate the time and effort the AC has devoted to reviewing our manuscript. To help the AC **quickly grasp** the core of our work and the key points of our rebuttal, we provide the following **brief summary**.

In this paper, we meticulously captured and manually selected high-quality images containing rich textures to construct a large-scale plant dataset, PlantRSR, comprising 16,585 HR–Ref pairs. The dataset captures the complexity and variability of plant scenes through extensive variations. In addition, we propose a novel RefSR method specifically designed to tackle the distinct challenges posed by plant imagery. It incorporates a Selective Key-Region Matching (SKRM) that selectively identifies and performs matching between LR and Ref images, focusing on distinctive botanical textures to improve matching efficiency. Additionally, a Texture-Guided Diffusion Module (TGDM) is proposed to refine LR textures by leveraging a diffusion process conditioned on the matched Ref textures. TGDM is effective in modeling irregular and fine textures, thereby facilitating more accurate SR results.

Before the discussion phase, we received two scores of 6, one score of 4, and one score of 2.

> Reviewer rAhJ (Score: 2 / Confidence: 4) had some **misunderstandings** about our key contributions, which we clarified thoroughly in the rebuttal. They also questioned the improvement brought by our dataset and suggested evaluating with additional settings and metrics. We conducted further experiments accordingly and added the results in the rebuttal. In addition, as requested, we corrected minor errors and improved citations in the main paper.

> Reviewer  tpSi (Score: 6 / Confidence: 4) asked for additional experiments and clarifications. Most of the required experiments were already included in the appendix, and we have incorporated the relevant results and explanations into the main text as requested.

> Reviewer gwnq (Score: 4 / Confidence: 4) was primarily concerned about the data collection process of our dataset. We provided more detailed categorization and acquisition descriptions in both the rebuttal and the revised main paper, and we also addressed the reviewer’s technical questions regarding the method.

> Reviewer VMqn (Score: 6 / Confidence: 4) raised multiple questions and engaged in several rounds of discussion during the discussion phase. We have addressed all concerns and updated the main paper accordingly based on the reviewer’s suggestions.

Following the reviewers’ suggestions and questions, we **revised** the manuscript accordingly. The major updates include: providing a more comprehensive set of references, correcting minor errors throughout the paper, and adding metric-based annotations to the visual comparison results. We also added Table 5, which reports experiments on the diffusion steps used in TGDM, as well as Tables 6 and 7, which offer a more detailed description of the dataset categories. We hope these revisions further improve the readability of our manuscript.

We would like to express our sincere gratitude to the reviewers and ACs once again for their thorough evaluation and insightful suggestions, which have significantly strengthened the quality of this work. We firmly believe that our dataset represents a substantial contribution to the community by addressing the scarcity of high-quality plant datasets, and our proposed method has demonstrated clear effectiveness across multiple settings. We sincerely hope that the AC will consider our work with these contributions in mind.

---

### Meta-Review · Area_Chair_CXLa · 2026-01-08

**Summary:**

This paper presents PlantRSR, a reference-based super-resolution framework designed for plant imagery. Its main contribution lies in the construction of a large-scale, high-quality plant dataset containing 16,585 HR Ref pairs and the proposal of a RefSR method that includes Selective Key Region Matching and a Texture Guided Diffusion Module. Reviewers generally agree that exploring RefSR within complex botanical textures is a highly practical and well-motivated topic that fills a gap in existing benchmarks. Overall, this work receives support due to its competitive performance and the specialized value of the proposed dataset.
However, reviewers also raise several concerns during the initial review. Main issues include limited technical novelty, concerns regarding the small improvements in global metrics, and the need for more rigorous evaluations of dataset diversity and downstream utility. In the rebuttal, the authors conduct extensive additional experiments based on Masked PSNR to isolate texture-dense regions, provide a detailed breakdown of plant categories and environmental factors, and illustrate the benefits of the framework for downstream plant classification tasks. The AC thinks that these responses effectively clarify the true impact of this work beyond global average metrics. Based on the above content, the AC believes that this paper meets the acceptance standards for the datasets and benchmarks area.

**Reviewer Concerns:**

**Concerns that are largely addressed in the rebuttal**

- The authors introduce M-PSNR and M-SSIM to evaluate performance specifically on texture-relevant regions. This addresses the initial concern regarding marginal global improvements and demonstrates that performance gains are substantial in areas where textures are actually present.
- The authors provide a comprehensive classification of plant species, growth stages, and environmental variations such as illumination, weather, and decay. This clarifies the scientific rigor of the dataset and resolves concerns about its robustness in real-world scenarios.
- The authors conduct a detailed trade-off analysis between performance and efficiency for the number of diffusion steps. They prove that the method choosing T=4 achieves a good balance between reconstruction quality and practical usability.
- The authors apply the model as a preprocessing step for plant species classification in PlantCLEF2022. The improvement in MA-MRR confirms the practical value of the dataset and method beyond serving as a standard benchmark.

**Concerns that remain unresolved**

- The authors explain the unique principles behind SKRM and TGDM to differentiate them from dense matching and diffusion priors. However, the AC tends to agree with the concern raised by reviewers and considers the method more of an incremental integration of existing components, such as block matching and diffusion priors.

**Reviewer Scores:**

**Reviewer rAhJ:** This reviewer likely raises the score to 5 after considering the rebuttal. The authors use M-PSNR to refute the claim of limited improvement, which removes the primary reasons for the original rejection.

**Reviewer tpSi:** The rebuttal addresses technical questions regarding real-world retrieval scenarios and efficiency. Since the authors provide comprehensive additional data, the reviewer is most likely to increase the score to 7.

**Reviewer gwnq** The authors clarify the adaptive mechanism of SKRM and provide the requested botanical taxonomy for the dataset. These clarifications align with the reviewer's requests for dataset supplements, and the score is likely to adjust to 6.

**Reviewer VMqn** This reviewer participates in the follow-up discussion and accepts the M-PSNR analysis. Although the reviewer maintains reservations about the downstream improvements, they remain positive about the value of the dataset. The AC thinks the score is likely to remain at 6.

---

### Decision · Program_Chairs · 2026-01-26

Accept (Poster)